# A Derandomization Framework for Structure Discovery: Applications in Neural Networks and Beyond

**Nikos Tsikouras**[1,2]   **Yorgos Pantis**[1,2]   **Ioannis Mitliagkas**[2,3]   **Christos Tzamos**[1,2]
[1] National and Kapodistrian University of Athens, Greece
[2] Archimedes, Athena Research Center, Greece
[3] Mila & Université de Montréal, Canada

## Abstract

Understanding the dynamics of feature learning in neural networks (NNs) remains a significant challenge. The work of (Mousavi-Hosseini et al., 2023) analyzes a multiple index teacher-student setting and shows that a two-layer student attains a low-rank structure in its first-layer weights when trained with stochastic gradient descent (SGD) and a strong regularizer. This structural property is known to reduce sample complexity of generalization. Indeed, in a second step, the same authors establish algorithm-specific learning guarantees under additional assumptions. In this paper, we focus exclusively on the structure discovery aspect and study it under weaker assumptions, more specifically: we allow (a) NNs of arbitrary size and depth, (b) with all parameters trainable, (c) under any smooth loss function, (d) tiny regularization, and (e) trained by any method that attains a second-order stationary point (SOSP), e.g. perturbed gradient descent (PGD). At the core of our approach is a key *derandomization* lemma, which states that optimizing the function $\mathbb{E}_{\boldsymbol{x}}\left[g_\theta(\boldsymbol{W}\boldsymbol{x} + \boldsymbol{b})\right]$ converges to a point where $\boldsymbol{W} = \boldsymbol{0}$, under mild conditions. The fundamental nature of this lemma directly explains structure discovery and has immediate applications in other domains including an end-to-end approximation for MAXCUT, and computing Johnson-Lindenstrauss embeddings.

## 1 Introduction

Neural networks (NNs) have become successful tools across different domains, demonstrating exceptional performance in complex tasks, such as image recognition, natural language processing, or speech synthesis (LeCun et al., 2015; Goodfellow, 2016). This broad applicability is primarily due to their ability to learn and generalize from large datasets, enabling them to identify challenging patterns and relationships that are difficult to capture with traditional techniques. Theoretical work in this area focuses on various aspects, including the structure of optimization landscapes, and generalization behavior, aiming to answer fundamental questions about why these models work as well as they do and how they can be made more efficient and trustworthy (Arora et al., 2017; Montavon et al., 2018; Neyshabur et al., 2019).

Since data is crucial in this line of research, *teacher models* have emerged in learning theory as a formalism for structured data. Extensive research has been conducted on this topic, particularly when the trained (*student*) model is a NN, offering precise and non-asymptotic guarantees in various contexts (Zhong et al., 2017; Goldt et al., 2019; Ba et al., 2020; Sarao Mannelli et al., 2020; Zhou et al., 2021; Akiyama & Suzuki, 2021; Abbe et al., 2022; Ba et al., 2022; Damian et al., 2022; Veiga et al., 2022; Mousavi-Hosseini et al., 2023). Experimental evidence suggests that traditional learning theory fails to fully explain the generalization properties of large NNs, highlighting the need for more modern approaches (Zhang et al., 2021).

---

Code available at https://github.com/TPMT26/StructureDiscovery

An important concept that frequently appears in modern learning theory is the implicit regularization effect introduced by training dynamics (Neyshabur et al., 2015a). The work of (Soudry et al., 2018) sparked a wave of recent studies investigating how gradient descent (GD) naturally tends to favor lower-complexity models, often leading to minimum-norm and/or maximum-margin solutions even without explicit regularization (Gunasekar et al., 2018; Li et al., 2018b; Ji & Telgarsky, 2019; Gidel et al., 2019; Chizat & Bach, 2020; Pesme et al., 2021). However, much of this research focuses on linear models or excessively wide NNs, with varying interpretations of reduced complexity and its impact on generalization. A notable example in this context is compressibility and its relationship to generalization (Arora et al., 2018; Suzuki et al., 2020). When a trained NN can be compressed into a smaller model with similar predictive behavior, both models show comparable generalization performance. This suggests that the original NN's complexity may be understood via its simpler compressed form, which is traditionally associated with improved generalization.

A key contribution in the area, and influence for our work, is the work of (Mousavi-Hosseini et al., 2023), which studies the training dynamics of a two-layer NN using stochastic gradient descent (SGD) on data drawn from a multiple-index teacher model. First, they show that low-complexity structures emerge during training when a strong regularizer is used: on first-order stationary points (FOSPs), the first layer weights align with key directions in the input space, the *principal subspace*. Low-dimensional structure of this type is known to help with generalization (Neyshabur et al., 2015b; Bartlett et al., 2017). As a second step, they establish GD-specific generalization guarantees under additional assumptions. Our focus is on the first step, and the goal of this work is to answer the question:

*Can we discover low-rank structure in neural networks under more natural assumptions?*

To this end, we consider a more precise and well-motivated solution concept, a $\rho$-approximate *second-order stationary point* ($\rho$-SOSP), (Jin et al., 2017), see Definition 2.2. Using the properties of $\rho$-SOSPs, we provide a general derandomization lemma. When applied specifically to NNs, our lemma implies that, for any *arbitrarily small* regularizer value, there exists a $\rho > 0$ such that all $\rho$-SOSPs correspond to low-rank first-layer weights. Thus, we significantly relax the sufficient conditions for uncovering this kind of structure as we allow (a) NNs of arbitrary size and depth, (b) with all parameters trainable (including biases), (c) under any smooth loss function, (d) arbitrarily small regularization, and (e) trained by any method that attains a $\rho$-SOSP, for example, perturbed gradient descent (PGD) and Hessian descent.

**Importance of training the biases.** For the sake of analytical simplicity, some past work has frozen the biases, for example in (Mousavi-Hosseini et al., 2023). Here, we show that training the biases is necessary for derandomization to take effect. To illustrate this point, we consider the following toy example. Let $x$ be a one-dimensional standard Gaussian random variable, and suppose the target label is fixed at 1, i.e., the output lies in a zero-dimensional space. We model the prediction using a single-layer NN. The objective is to minimize the loss function:

$$f(w, b) = \mathbb{E}\left[\left(\text{ReLU}^3(wx + b) - 1\right)^2\right] + \lambda w^2,$$

where $w$ and $b$ denote the weight and bias parameters, respectively, and $\lambda > 0$ is the regularizer. Directly applying the result from (Mousavi-Hosseini et al., 2023), implies that $w = 0$, i.e. the solution lies in a zero-dimensional space. However, this solution is evidently suboptimal when $b \neq 1$. Consequently, enforcing this behavior requires an artificially large regularization $\lambda$. We illustrate this phenomenon in Figure 1 where the minimizer $w^*$ approaches zero only under large values of $\lambda$.

In contrast, our analytical approach, which allows the training of biases, avoids this drawback by considering $\rho$-SOSPs. Instead of requiring an artificially large regularization parameter to explain this structural behavior ($w = 0$), we only need a tiny amount of regularization, as shown in Figure 2, because the bias term can adjust to $b = 1$. This shows that freezing the biases places an unnecessary restriction, whereas allowing them to be trained explains the phenomenon more directly and under milder conditions.

**Discussion on $\rho$-SOSPs.** Training the biases allows for smaller regularizers, but proving low-rank solutions remains challenging. In (Mousavi-Hosseini et al., 2023), a strong regularizer is used to show

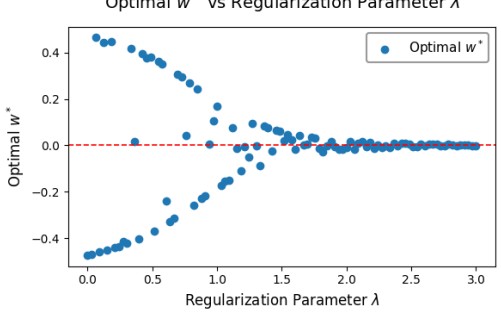
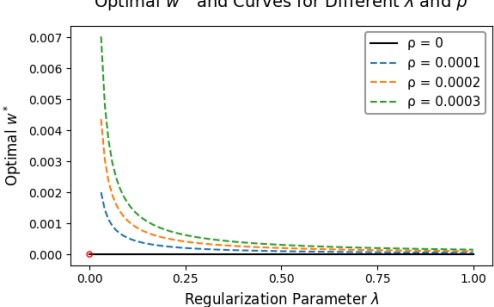

Figure 1: Plot of the global minimizer of $f(w, 0) = \mathbb{E}[(\text{ReLU}^3(wx) - 1)^2] + \lambda w^2$ as a function of the regularization parameter $\lambda$.

Figure 2: Optimal $w$ vs. $\lambda$ for $f(w, b) = \mathbb{E}[(\text{ReLU}^3(wx + b) - 1)^2] + \lambda w^2$ when biases are trained. The solid line shows convergence to 0 at an exact SOSP.

that no FOSP can be high-rank; without it, higher-rank FOSPs may exist. We address this analytical challenge by exploiting the extra properties of $\rho$-SOSPs, which bound the negative curvature. The $\rho$-SOSP solution concept excludes all but the flattest of saddle points, in other words, a $\rho$-SOSP is more likely to be a local minimum than the corresponding approximate FOSP. Using this, we show that for sufficiently small $\rho$, the only valid solutions are low-rank.

We remark that our focus is on the *landscape* of the objective (Equation 1), not specific optimization methods. Importantly, $\rho$-SOSPs capture the solutions reached by standard algorithms: GD almost surely avoids strict saddles (Panageas et al., 2019), and PGD (Jin et al., 2017) guarantees efficient convergence to $\rho$-SOSPs. Crucially, all local minima are $\rho$-SOSPs. Furthermore, empirical observations show that gradient-based methods reliably reach good minima while avoiding saddles (Li et al., 2018a; Zhou et al., 2020) and that these solutions have good learning properties. Thus, the focus on $\rho$-SOSPs is well motivated as it captures the types of solutions seen in practice and theory, while also providing a framework to study the mechanisms of implicit regularization.

**Summary of our contributions.** Our key contributions can be summarized as follows:

We consider a general family of functions of the form:

$$f(\boldsymbol{W}, \boldsymbol{b}; \theta) = \mathbb{E}_{\boldsymbol{x}} \left[ g_\theta(\boldsymbol{W}\boldsymbol{x} + \boldsymbol{b}) \right] + \lambda \|\boldsymbol{W}\|_F^2, \tag{1}$$

where $\boldsymbol{W} \in \mathbb{R}^{k \times d}$, $\boldsymbol{b} \in \mathbb{R}^k$, $\boldsymbol{x} \sim \mathcal{N}(\boldsymbol{0}, \boldsymbol{I}_d)$, $g_\theta(\cdot) : \mathbb{R}^k \to \mathbb{R}$ denotes a parameterized nonlinear function and $\lambda > 0$ is a regularization parameter. This formulation is highly general and encompasses a wide range of applications, including the population risk of NNs of arbitrary depth and architecture, under any smooth loss functions.

Below we present our key *derandomization* lemma, which forms the foundation of our results, stated informally as follows:

**Informal version of Lemma 3.1** Let $f$ be a twice differentiable function in the form of Equation 1, where $\boldsymbol{x} \sim \mathcal{N}(\boldsymbol{0}, \boldsymbol{I}_d)$. Then, all SOSPs of $f$ satisfy $\boldsymbol{W} = \boldsymbol{0}$.

**Structure discovery in NNs.** We first show that the regularized risk of any NN can be expressed as a function of the form in Equation 1. Then, by applying our key *derandomization* lemma, we establish that in the teacher-student setting any $\rho$-SOSP solution of the risk yields a first-layer weight matrix $\boldsymbol{W}$ which is near low-rank. This type of structure is closely associated with improved generalization performance as shown in (Mousavi-Hosseini et al., 2023). In this paper, our primary focus is on providing guarantees for structure discovery under much broader and more minimal assumptions, thereby capturing a wider class of models and cases of arbitrarily weak regularization.

In other words, our main result as applied to NNs suggests a new explanation for parsimony even with weak regularizers. Recall that standard methods like SGD, GD, or PGD are known to escape saddle points; when that happens, our results guarantee a solution with good structure (i.e. low rank).

**Our results in other domains.** Due to the generality of our *derandomization* lemma we obtain strong theoretical results across domains. For example, we get (i) a deterministic MAXCUT approximation matching the randomized guarantee of (Goemans & Williamson, 1995), and (ii) a deterministic construction for learning Johnson-Lindenstrauss (JL) embeddings (Johnson et al., 1984). In the case of MAXCUT, our contribution is to show that derandomization can be achieved via simple gradient-based optimization, without relying on explicit combinatorial constructions or pseudorandom generators. To the best of our knowledge, this is the first optimization-based approach for derandomizing the Goemans-Williamson algorithm, and we view it as a conceptual contribution that enriches the literature on derandomization. In the case of JL, we match the state-of-the-art result of (Tsikouras et al., 2024) further demonstrating the generality of our lemma. We expect the same approach to extend to other domains.

## 2 NOTATION AND PRELIMINARIES

**Notation.** For vectors $\boldsymbol{u}, \boldsymbol{v}$ we use $\langle \boldsymbol{u}, \boldsymbol{v} \rangle$ or $\boldsymbol{u} \cdot \boldsymbol{v}$ to denote their inner product and $\|\boldsymbol{u}\|_2$ to denote the $L_2$ norm. For matrix $\boldsymbol{M} \in \mathbb{R}^{k \times d}$, we denote the element of the $i^{th}$ row and $j^{th}$ column by $\mu_{i,j}$ and we use $\|\boldsymbol{M}\|_F$ to denote the Frobenius norm. We use $\nabla f$ and $\nabla^2 f$ to denote the gradient and Hessian operators, respectively. Additionally, we use $\mathbf{A} \sim \mathcal{N}(\boldsymbol{M}, \boldsymbol{\Sigma})$, where $\boldsymbol{M} = (\mu_{i,j})$ and $\boldsymbol{\Sigma} = (\sigma_{i,j})$ to indicate that $\mathbf{A} = (a_{i,j})$ is a matrix with independent random entries, and each entry follows $a_{i,j} \sim N(\mu_{i,j}, \sigma_{i,j})$. Using this notation we can write $\mathbf{A} = \boldsymbol{M} + \mathbf{Z}$, where $\mathbf{Z} \sim \mathcal{N}(\mathbf{0}, \boldsymbol{\Sigma})$.

**Definition 2.1.** *A twice differentiable function $f : \mathbb{R}^d \to \mathbb{R}$ is defined to be $L$-smooth, if for all $\boldsymbol{x}, \boldsymbol{y} \in \mathbb{R}^d$ it satisfies:*

$$\|\nabla f(\boldsymbol{x}) - \nabla f(\boldsymbol{y})\|_2 \le L\|\boldsymbol{x} - \boldsymbol{y}\|_2.$$

*The function is $K$-Hessian Lipschitz if for all $\boldsymbol{x}, \boldsymbol{y} \in \mathbb{R}^d$:*

$$\|\nabla^2 f(\boldsymbol{x}) - \nabla^2 f(\boldsymbol{y})\|_2 \le K\|\boldsymbol{x} - \boldsymbol{y}\|_2.$$

Below, we give the definition for approximate stationarity.

**Definition 2.2** (Approximate second-order stationarity). *For a $K$-Hessian Lipschitz function $f(\cdot)$, we say that a point $\boldsymbol{x}^*$ is a $\rho$-second-order stationary point ($\rho$-SOSP) if:*

$$\|\nabla f(\boldsymbol{x}^*)\|_2 \le \rho \quad and \quad \lambda_{\min}(\nabla^2 f(\boldsymbol{x}^*)) \ge -\sqrt{K\rho}.$$

**Assumption 2.3.** *The function is both $L$-smooth and $K$-Hessian Lipschitz.*

Provided Assumption 2.3 holds, Algorithm 1 converges to a $\rho$-SOSP in $\mathcal{O}(1/\rho^2)$ iterations with high probability (Jin et al., 2017). Alternatively, Algorithm 2 achieves a deterministic $\rho$-SOSP in $\mathcal{O}(1/\rho^{1.5})$ iterations (Tsikouras et al., 2024), but requires Hessian access.

## 3 MAIN CONTRIBUTION: KEY DERANDOMIZATION LEMMA

In this section, we prove that convergence to SOSPs is a sufficient condition for derandomization. Let $\boldsymbol{W} \in \mathbb{R}^{k \times d}, \boldsymbol{b} \in \mathbb{R}^k, g_\theta(\cdot) : \mathbb{R}^k \to \mathbb{R}$ be a function satisfying Assumption 2.3, and let $\boldsymbol{x} \sim \mathcal{N}(\mathbf{0}, \boldsymbol{I}_d)$. We analyze the behavior of the following objective function at its SOSPs:

$$f(\boldsymbol{W}, \boldsymbol{b}; \theta) = \mathbb{E}_{\boldsymbol{x}}[g_\theta(\boldsymbol{W}\boldsymbol{x} + \boldsymbol{b})] + \lambda\|\boldsymbol{W}\|_F^2, \tag{2}$$

where $\lambda > 0$ is an arbitrarily small regularization parameter. Our main result is presented below.

**Lemma 3.1** (Key Derandomization Lemma). *Let $\boldsymbol{x} \sim \mathcal{N}(\mathbf{0}, \boldsymbol{I}_d)$ be a standard multivariate Gaussian random variable. For the objective function defined in Equation 2, with $\lambda > \frac{\sqrt{K\rho}}{2}$ where $g_\theta(\cdot)$ satisfies Assumption 2.3, any $\rho$-SOSP satisfies $\|\boldsymbol{W}\|_F \le \frac{\rho}{2\lambda - \sqrt{K\rho}}$.*

**Proof sketch:** The key observation is that applying Stein's Lemma (Stein, 1973; 1981) relates the second derivative of $\boldsymbol{b}$ with the first derivative of $\boldsymbol{W}$. This allows us to express the first-order conditions for $\boldsymbol{W}$ in terms of expectations involving the second derivatives of $g_\theta$. Using these relations and an approximate first-order optimality condition, we derive the required bound on the Frobenius norm of $\boldsymbol{W}$. Full proof can be found in Appendix B.1. □

**Remark 3.2.** *Note that achieving a perfect SOSP (i.e.,$\rho = 0$), would result in $\boldsymbol{W} = \boldsymbol{0}$; the proof of this is in Appendix B.2.*

This result shows that when the objective function takes the form given in Equation 2, a common structure in practice, it is possible to improve it by minimizing the inherent randomness. Since the input $\boldsymbol{x}$ is random, having a small $\|\boldsymbol{W}\|_F$, ensures that $\boldsymbol{W}\boldsymbol{x}$ remains small on average, so the values of $g_\theta(\boldsymbol{W}\boldsymbol{x} + \boldsymbol{b})$ vary less. In other words, second-order stationarity implies that the randomness in the objective function is effectively decreased. We illustrate the implications of this result by applying it to three distinct examples from different fields, as demonstrated in the following sections. We emphasize that the $\rho$-SOSP is taken with respect to all model parameters.

Additionally, we clarify that our analysis operates under the assumption that the optimization method employed converges to a $\rho$-SOSP with respect to all model parameters, including those encapsulated in $\theta$. Specifically, we assume that an approximate SOSP is identified jointly over the entire parameter space. Once such a point is found, we observe that fixing the auxiliary parameters $\theta$ preserves the approximate second-order stationarity with respect to the first-layer weight matrix $\boldsymbol{W}$.

It is important to note that a tiny amount of regularization is necessary to ensure a well-defined solution in this lemma. Without it, choosing the function $g_\theta(wx + b) = 0$, would result in all $w \in \mathbb{R}$ being local minima, as the objective provides no preference among these values. Introducing a tiny regularization term $\lambda$ eliminates this ambiguity by penalizing non-zero weights, thereby enforcing $w = 0$ as the unique optimal solution.

Additionally, $\lambda$ is fully controllable by $\rho$, which is set prior to the optimization process. As a result, it is possible to use arbitrarily small regularization, provided one is willing to incur the additional cost of executing the optimization algorithm for a greater number of steps. We emphasize this point to remind the reader that only a minimal amount of regularization is necessary. Therefore, the regularization term itself is not the primary factor influencing the outcome; rather, it is the interaction between the second derivative of $\boldsymbol{b}$ and the first derivative of $\boldsymbol{W}$ that plays a more significant role in the optimization process.

It is important to clarify that Lemma 3.1 serves as a *structure discovery* result rather than an *optimization* result. In particular, our focus lies not on the specific optimization algorithm employed, but rather on establishing that any solution satisfying the $\rho$-SOSP condition necessarily reveals structure. There might be different methods under many different sets of assumptions that yield a $\rho$-SOSP solution, even in more practical and realistic finite-sample regimes. Nonetheless, for completeness, we note that PGD (Jin et al., 2017), originally developed for deterministic objectives such as the population risk, has been shown to efficiently yield $\rho$-SOSP solutions in polynomial time in stochastic and finite-sample regimes (see Theorem 15 in (Jin et al., 2018)).

## 4   STRUCTURE DISCOVERY IN NEURAL NETWORKS

In this section, we present our primary application, which builds on the central *derandomization* Lemma 3.1. Our approach extends the work of (Mousavi-Hosseini et al., 2023), which demonstrated a key insight: the convergence of the first-layer weights to a low-dimensional subspace. We refine and generalize these results to a broader setting. Let $\boldsymbol{x} \in \mathbb{R}^d$, be a standard Gaussian distribution $\boldsymbol{x} \sim \mathcal{N}(\boldsymbol{0}, \boldsymbol{I}_d)$. The target labels are generated by a multiple-index teacher model of the form:

$$y = h(\langle \boldsymbol{u}_1, \boldsymbol{x} \rangle, \ldots, \langle \boldsymbol{u}_k, \boldsymbol{x} \rangle; \epsilon) \equiv h(\boldsymbol{U}\boldsymbol{x}; \epsilon), \tag{3}$$

where $h : \mathbb{R}^{k+1} \to \mathbb{R}$ is a weakly differentiable link function and $\epsilon$ represents additive noise.

For any vector $\boldsymbol{v} \in \mathbb{R}^d$, let $\boldsymbol{v}_\|$ denote its orthogonal projection onto $\mathrm{span}(\boldsymbol{u}_1, \ldots, \boldsymbol{u}_k)$, and define $\boldsymbol{v}_\perp := \boldsymbol{v} - \boldsymbol{v}_\|$. For a matrix $\boldsymbol{W} \in \mathbb{R}^{k \times d}$, we define $\boldsymbol{W}_\|$ and $\boldsymbol{W}_\perp$ by projecting each row of $\boldsymbol{W}$ similarly. Using this notation, we rewrite the labeling function to depend only on the $\boldsymbol{x}_\|$ component:

$$y = h(\boldsymbol{U}\boldsymbol{x}) = h'(\boldsymbol{x}_\|).$$

Our goal is to show that the perpendicular component $\boldsymbol{W}_\perp$ converges to zero, implying that the first-layer weight matrix $\boldsymbol{W}$ lies entirely in the teacher subspace.

(Mousavi-Hosseini et al., 2023) showed that, under certain conditions, training only the first-layer weights of a two-layer NN suffices to ensure convergence to the low-dimensional principal subspace defined by the teacher model. We extend this result by showing that training the first-layer bias is also essential. Including the bias enables a simpler and more direct analysis of the training dynamics.

This expanded approach allows us to establish convergence guarantees under significantly more general conditions, including: (a) arbitrary regularization parameters $\lambda > 0$ (resolving an open question in earlier work), (b) arbitrary NN size and depth, (c) all parameters being trainable (including biases), and (d) any choice of smooth loss function.

## 4.1 Discovering structure in neural networks via SOSPs

Let $\boldsymbol{W} \in \mathbb{R}^{k \times d}$ and $\boldsymbol{b} \in \mathbb{R}^k$, and consider the first layer of a NN given by $\boldsymbol{W}\boldsymbol{x} + \boldsymbol{b}$. We decompose this expression into components that are parallel and perpendicular to a subspace $U$, as follows:

$$\boldsymbol{W}\boldsymbol{x} + \boldsymbol{b} = \boldsymbol{W}_\parallel \boldsymbol{x}_\parallel + \boldsymbol{W}_\perp \boldsymbol{x}_\perp + \boldsymbol{b}, \tag{4}$$

since $\boldsymbol{W}_\perp \boldsymbol{x}_\parallel = \boldsymbol{0}$ and $\boldsymbol{W}_\parallel \boldsymbol{x}_\perp = \boldsymbol{0}$ due to orthogonality.

Now define the NN's prediction as $\hat{y}(\boldsymbol{x}; \boldsymbol{W}, \boldsymbol{b}, \theta) = g_\theta(\boldsymbol{W}\boldsymbol{x} + \boldsymbol{b})$, where $g_\theta(\cdot)$ is a NN of arbitrary size and depth, parameterized by $\theta$, which satisfies Assumption 2.3. Given a loss function $\ell(y, \hat{y})$ that also satisfies Assumption 2.3, and a regularization parameter $\lambda > 0$, we define the regularized risk as:

$$R(\boldsymbol{W}, \boldsymbol{b}; \theta) := \mathbb{E}_{\boldsymbol{x}} \left[ \ell\left(y, \hat{y}(\boldsymbol{x}; \boldsymbol{W}, \boldsymbol{b}, \theta)\right) \right] + \lambda \|\boldsymbol{W}\|_F^2. \tag{5}$$

As shown in Appendix D, Equation 5 can be reformulated in a way that enables direct application of Lemma 3.1. In particular, this reformulation expresses the regularized risk as a function of the perpendicular components:

$$R(\boldsymbol{W}_\perp, \boldsymbol{b}; \theta') = \mathbb{E}_{\boldsymbol{x}_\perp} \left[ \ell'_{\theta'}\left(\boldsymbol{W}_\perp \boldsymbol{x}_\perp + \boldsymbol{b}\right) \right] + \lambda \|\boldsymbol{W}_\perp\|_F^2. \tag{6}$$

In this form, the parallel and perpendicular components are fully decoupled. The modified loss $\ell'_{\theta'}$ implicitly depends on the parallel components, the corresponding regularization, and all other parameters of the NN. By applying Lemma 3.1, we conclude that $\|\boldsymbol{W}_\perp\|_F$ can be made arbitrarily small, implying that the perpendicular components are effectively suppressed during training.

**Theorem 4.1.** *Consider an arbitrary NN of any size and depth that satisfies Assumption 2.3, and a loss function that also satisfies this assumption. Let the input data $\boldsymbol{x} \sim \mathcal{N}(\boldsymbol{0}, \boldsymbol{I}_d)$ be a standard multivariate Gaussian distribution, and the labels generated according to Equation 3. If we minimize Equation 6 with respect to $(\boldsymbol{W}, \boldsymbol{b})$, then for any precision parameter $\rho$ and regularization parameter $\lambda > \frac{\sqrt{K\rho}}{2}$, where $K$ denotes the Hessian Lipschitz constant of the objective, all $\rho$-SOSPs satisfy the inequality:*

$$\|\boldsymbol{W}_\perp\|_F \leq \frac{\rho}{2\lambda - \sqrt{K\rho}}.$$

**Proof.** Since both the NN and the loss function are smooth and Hessian Lipschitz, their composition inherits these properties. The result then follows directly from Lemma 3.1. □

The result establishes the theoretical validity of our approach but is qualitative in nature, as it does not specify the number of steps required for optimization. To complement this, we provide a quantitative result demonstrating that the objective function can be minimized efficiently.

**Theorem 4.2.** *Let the objective function in Equation 5 be twice differentiable, bounded below, and satisfies Assumption 2.3. Let the data $\boldsymbol{x} \sim \mathcal{N}(\boldsymbol{0}, \boldsymbol{I}_d)$, and suppose labels are generated according to Equation 3. Let $\rho > 0$ be a prespecified accuracy, and define the regularization parameter $\lambda = \frac{\sqrt{K\rho} + \Delta}{2}$, where $K$ is the Hessian Lipschitz constant of the objective and $\Delta > 0$ is an arbitrarily small constant. Then, with probability $1 - \delta$, running Algorithm 1 for $T > \mathcal{O}\left(\text{poly}\left(L, \log(d), \log(\delta), \varepsilon^{-1}, \Delta^{-1}\right)\right)$ iterations with a step size $\mathcal{O}(1/L)$, yields a weight matrix $\boldsymbol{W} = \boldsymbol{W}_\perp + \boldsymbol{W}_\parallel$ that satisfies:*

$$\|\boldsymbol{W}_\perp\|_F < \varepsilon.$$

**Proof.**    Full proof can be found in Appendix E.1.                                      □

This result demonstrates that, with a number of samples that scales with the Hessian Lipschitz constant, PGD can effectively generate iterates that are as close to the principal subspace as required. This allows the model to learn low-dimensional representations, and introduces an implicit bias toward simpler, lower-complexity solutions. The discovery of structure appears to be an inherent characteristic of this optimization process for problems of this nature. This convergence to a low-dimensional solution is often linked with the generalization behavior of NNs (Neyshabur et al., 2015b; Bartlett et al., 2017; Arora et al., 2018; Suzuki et al., 2020; Mousavi-Hosseini et al., 2023). We do not attempt to establish generalization guarantees here, as such results would require additional assumptions on the data distribution or the hypothesis class. Instead, our contribution is to provide guarantees for structure discovery under broader and more minimal conditions, thereby extending the potential applicability of these ideas to a wider range of models.

Our results pertain to NNs with smooth activation functions; here we discuss the wide practical applicability of this family of networks. Smooth nonlinearities are widely used in modern architectures, and there is no strong evidence that non-smooth activations outperform their smooth counterparts. For example, BERT adopts the Gaussian error linear unit (GELU) (Hendrycks & Gimpel, 2016; Devlin et al., 2019), a smooth activation that has been shown to benefit from this choice compared with non-smooth alternatives. Thus, our assumption is aligned with standard practice. We provide additional insights for the non-smooth ReLU case by employing a smooth approximation in the next section.

### 4.2    THE CASE FOR ReLU

The non-smoothness of ReLU poses challenges for our framework. To ensure the smoothness and Hessian Lipschitz continuity needed for defining $\rho$-SOSPs, we use a smooth approximation of ReLU:

$$\text{ReLU}_\iota(x) = \frac{1}{\iota} \log\left(1 + e^{\iota x}\right).$$

As $\iota \to \infty$, the function converges to the standard ReLU. Moreover, it is $\frac{\iota}{4}$-gradient Lipschitz and $\frac{\sqrt{3}\iota^2}{9}$-Hessian Lipschitz, ensuring that our framework remains valid for any smooth ReLU approximation. In this sense, $\text{ReLU}_\iota$ captures the essential behavior of ReLU while enabling theoretical guarantees. To show the dependence on $\iota$ we give the following theorem, regarding a one layer NN using activation function $\text{ReLU}_\iota(\cdot)$.

**Theorem 4.3.** *Assume that the data $\boldsymbol{x} \sim \mathcal{N}(\boldsymbol{0}, \boldsymbol{I}_d)$, and labels are generated according to Equation 3. Additionally, consider the NN $\boldsymbol{a}^\top \text{ReLU}_\iota(\boldsymbol{W}\boldsymbol{x} + \boldsymbol{b})$ and the objective in Equation 5, which is twice differentiable, bounded below, and satisfies Assumption 2.3, with gradient and Hessian Lipschitz constants $L_\ell$ and $K_\ell$, respectively. Let $\rho > 0$ be a prespecified accuracy and define the regularization parameter $\lambda = \frac{\sqrt{K\rho} + \Delta}{2}$, where $K = \mathcal{O}(\iota^2 K_\ell)$ is the overall Hessian Lipschitz constant and $\Delta > 0$ is arbitrarily small. Then, with probability $1 - \delta$, running Algorithm 1 for $T > \mathcal{O}\left(\text{poly}\left(\iota, L_\ell, \log(\delta), \varepsilon^{-1}, \Delta^{-1}\right)\right)$ iterations, with a step size $\mathcal{O}(1/(\iota L_\ell))$, yields a weight matrix $\boldsymbol{W} = \boldsymbol{W}_\perp + \boldsymbol{W}_\parallel$ that satisfies:*

$$\|\boldsymbol{W}_\perp\|_F < \varepsilon.$$

**Proof.**    This is a direct application of Theorem 4.2. The composition of the objective with $\text{ReLU}_\iota$ satisfies Assumption 2.3 and is lower bounded.                                      □

### 4.3    NEURAL NETWORKS EXPERIMENTS

In this section, we empirically validate our theoretical framework by showing that the first layer weight matrix of the student network converges to the principal subspace. The teacher network is a single-index model that generates outputs according to $y = \tanh(\theta \cdot \boldsymbol{x}) + \text{noise}$, where $\boldsymbol{x} \in \mathbb{R}^2$ and $\theta = \frac{1}{\sqrt{2}}(1, 1)^\top$ is a fixed direction. The student network attempts to learn this mapping using a two-layer NN of the form

$$y = \boldsymbol{a}^\top \text{ReLU}_2(\boldsymbol{W}\boldsymbol{x} + \boldsymbol{b}),$$

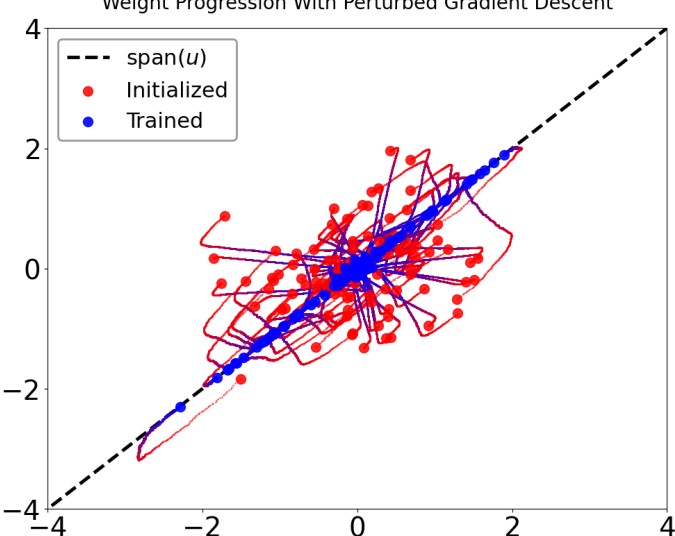

Figure 3: Two-layer $\text{ReLU}_2$ network of width $h = 1,000$ and $d = 2$ for the task of recovering a $\tanh$ single-index teacher model. We observe convergence of weights $\boldsymbol{W}$ to the principal subspace.

where $\boldsymbol{W} \in \mathbb{R}^{h \times d}$, $\boldsymbol{b} \in \mathbb{R}^h$, and $\boldsymbol{a} \in \mathbb{R}^h$. All parameters are randomly initialized, except that the second layer weights $\boldsymbol{a}$ are kept fixed throughout training to isolate the representation learning dynamics of the first layer. Full details can be found in Appendix F.

Empirically, we observe that the initially random rows of $\boldsymbol{W}$ align with the teacher direction and converge to the principal subspace spanned by $\theta$, as illustrated in Figure 3. This confirms that gradient-based optimization recovers the underlying signal structure despite random initialization. Moreover, we state for completeness, as rigorously established in (Mousavi-Hosseini et al., 2023), the emergence of such low-rank structure reduces the effective dependence of generalization error on the ambient input dimension $d$.

Training is conducted using PGD. At each iteration, if the gradient norm falls below the threshold $\epsilon = 10^{-6}$, isotropic Gaussian noise is injected into the parameters. Concretely, for the first-layer weights,

$$\boldsymbol{W} \leftarrow \boldsymbol{W} + \delta \cdot \mathcal{N}(0, I_d),$$

with $\delta = 0.005$ analogous perturbations are applied to the bias terms $b$. Although this threshold is small, the noise is applied many times during training, ensuring exploration of flat regions of the loss landscape. This modification of standard SGD helps the optimization escape saddle points and flat regions, which is particularly useful in our setting and aligns with our theoretical guarantees.

Training proceeds for $T = 10,000$ steps, minimizing the mean squared error (MSE) loss function with $L_2$ regularization applied only to the first layer weights $\boldsymbol{W}$ with coefficient $\lambda = 10^{-5}$. Learning rates are set to $\eta = 1$ for $\boldsymbol{W}$ and $\eta_b = 1$ for $\boldsymbol{b}$.

## 5 OTHER APPLICATIONS

### 5.1 MAXCUT

The MAXCUT problem is a classical combinatorial optimization problem that seeks to partition the vertices of a graph $G = (V, E)$ into two disjoint sets, $S$ and $T$, such that the sum of the weights of the edges crossing between $S$ and $T$ is maximized. The objective function for the MAXCUT problem is:

$$\text{Maximize:} \quad \sum_{(i,j) \in E} \frac{w_{ij}(1 - x_i x_j)}{2}, \quad \text{subject to} \quad x_i \in \{-1, 1\}, \quad \forall i \in V.$$

where $w_{ij}$ denotes the weight of the edge $(i,j)$ and the term $1 - x_i x_j$ equals 1 if edge $(i,j)$ crosses the cut and 0 otherwise. We follow the common assumption that $w_{i,j} = 1$. This is a combinatorial optimization problem that is NP-complete (Karp, 1972).

To make the problem more tractable, (Goemans & Williamson, 1995) proposed using a semidefinite program (SDP) relaxation. In this relaxation, the discrete MAXCUT problem is lifted to a continuous one by representing each vertex $i$ as a unit vector $\boldsymbol{v}_i \in \mathbb{R}^m$ on the unit sphere, where $m$ is the number of nodes in the graph. The algorithm first solves the SDP to obtain these vectors. It then applies a *randomized rounding* procedure to map the continuous solution back to a discrete cut: a standard Gaussian vector $\boldsymbol{z} \sim \mathcal{N}(\boldsymbol{0}, \boldsymbol{I}_m)$ is sampled, and each vertex is assigned to one of the two sets $S$ or $T$ based on the sign of the inner product $\langle \boldsymbol{v}_i, \boldsymbol{z} \rangle$.

We aim to approximate the MAXCUT problem by derandomizing this rounding algorithm. Let $\boldsymbol{V} \in \mathbb{R}^{m \times m}$ be the matrix of SDP vectors. Define $\boldsymbol{V}\boldsymbol{z} + \boldsymbol{\mu}$, where $\boldsymbol{z} \sim \mathcal{N}(\boldsymbol{0}, \boldsymbol{I}_m)$ and $\boldsymbol{\mu} \in \mathbb{R}^m$ is a mean vector. Our goal is to minimize the negative expected cut value, leading to the regularized objective function:

$$f(\boldsymbol{V}, \boldsymbol{\mu}) = -\sum_{i<j} w_{i,j} \Pr\left[\operatorname{sgn}(\boldsymbol{v}_i \cdot \boldsymbol{z} + \mu_i) \neq \operatorname{sgn}(\boldsymbol{v}_j \cdot \boldsymbol{z} + \mu_j)\right] + \lambda \|\boldsymbol{V}\|_F^2. \tag{7}$$

**Remark 5.1.** *The probability term in Equation 7 can be interpreted as the expectation of an indicator function for the event inside the probability. To allow the application of Lemma 3.1, we replace this indicator function with a smooth $\epsilon$-approximation, as described in Appendix G.1.*

We now present our main result on the derandomization of the randomized MAXCUT algorithm.

**Theorem 5.2** (Derandomized Approximation for MAXCUT). *Let $G = (V, E)$ be a graph with $m$ edges, where the edge weights are given by $w_{i,j} = w_{j,i} = 1$ for all $(i,j) \in E$. Let $\boldsymbol{V} \in \mathbb{R}^{m \times m}$ be the matrix of vectors obtained from the SDP relaxation of the MAXCUT problem, as described in (Goemans & Williamson, 1995). Denote by $\boldsymbol{z} \sim \mathcal{N}(\boldsymbol{0}, \boldsymbol{I}_m)$ a standard multivariate Gaussian vector, and let $\boldsymbol{\mu}$ represent a vector of means. Initialize Algorithm 1 with $\boldsymbol{\mu} = \boldsymbol{0}$, and optimize the $\epsilon$-smoothed version of the objective function in Equation 7 (see Equation 27), using the regularization parameter $\lambda = \frac{\sqrt{\rho/\epsilon^3} + \Delta}{2}$, and run for $T = \mathcal{O}\left(\operatorname{poly}\left(\log(m), \log(\delta), \epsilon^{-1}, \Delta^{-1}\right)\right)$ iterations. After this optimization process, the resulting vector $\boldsymbol{\mu}$ defines a cut whose value is guaranteed to be at least:*

$$\mathrm{OPT}(\alpha - \mathcal{O}(\epsilon)),$$

*with probability $1 - \delta$. Here, $\alpha = 0.878$ is the approximation factor from (Goemans & Williamson, 1995).*

**Proof.**  Full proof can be found in Appendix G.1. □

To the best of our knowledge, this work is the first optimization-based derandomization of the MAXCUT problem: rather than relying on the method of conditional expectations, small-bias spaces, or explicit pseudorandom constructions (Naor & Naor, 1990; Motwani & Raghavan, 1995; Mahajan & Ramesh, 1995). In addition, we have conducted experiments demonstrating our optimization-based construction for solving MAXCUT. Full details are provided in Appendix G.2.

## 5.2 JOHNSON-LINDENSTRAUSS EMBEDDINGS

The JL Lemma is a well-known result in the field of dimensionality reduction (Johnson et al., 1984). Specifically, consider unit norm data points $\boldsymbol{x}_1, \ldots, \boldsymbol{x}_n \in \mathbb{R}^d$, which we aim to project into $k$ dimensions while preserving their norms with at most $\varepsilon$-distortion. Here, the distortion is given by $\varepsilon = \mathcal{O}\left(\sqrt{\log n / k}\right)$. A detailed description of the JL Lemma is given in Appendix H.1. In this context, we are interested in finding matrices that satisfy the *JL guarantee*:

**Definition 5.3** (JL guarantee). *The JL guarantee states that for given dataset $\boldsymbol{x}_1, \ldots, \boldsymbol{x}_n \in \mathbb{R}^d$ and target dimension $k$, the distortion for all points does not exceed $\mathcal{O}(\sqrt{\log n / k})$.*

Significant research has been dedicated to improving the construction of random projections (Indyk & Motwani, 1998; Achlioptas, 2001; Matoušek, 2008). In contrast to these traditional methods, our

approach recovers the result from (Tsikouras et al., 2024), which proposes learning the linear mapping directly from the data, deterministically. Other derandomization methods for JL include (Engebretsen et al., 2002; Meka & Zuckerman, 2010).

Let $\mathbf{A}$ be a random matrix whose entries $a_{i,j}$ are independently drawn from a Gaussian distribution with means $\mu_{i,j}$ and variances $\sigma_{i,j}^2$. Let $\mathbf{\Sigma}$ denote the matrix collecting these variances. Our goal is to minimize the following quantity:

$$\Pr\left(\max_{i=1,\ldots,n}\left|\|\mathbf{A}\boldsymbol{x}_i\|_2^2 - 1\right| > \varepsilon\right) + \frac{\|\mathbf{\Sigma}^{1/2}\|_F^2}{2kd}, \tag{8}$$

which represents the probability that the maximum distortion across all input vectors exceeds a prescribed threshold $\varepsilon$, augmented by a regularization term that penalizes large variances. Notably, the regularizer vanishes as $\mathbf{\Sigma} \to \mathbf{0}$, recovering a deterministic transformation in the limit.

As shown in Appendix H.1, we use a union bound to relax the original objective in Equation 8, reducing it to an equivalent surrogate objective:

$$f\left(\mathbf{\Sigma}^{1/2}, \boldsymbol{\mu}\right) = \sum_{i=1}^{n} \Pr\left(\left|\left\|\left(\mathbf{\Sigma}^{1/2}\boldsymbol{z} + \boldsymbol{\mu}\right)\boldsymbol{x}_i\right\|_2^2 - 1\right| > \varepsilon\right) + \frac{\left\|\mathbf{\Sigma}^{1/2}\right\|_F^2}{2kd}, \tag{9}$$

where $\boldsymbol{z} \sim \mathcal{N}(\mathbf{0}, \boldsymbol{I}_{kd})$. The optimization is performed over the parameters $(\mathbf{\Sigma}^{1/2}, \boldsymbol{\mu})$.

**Remark 5.4.** *The probability term in Equation 9 can be interpreted as the expectation of an indicator function for the event inside the probability. To allow the application of Lemma 3.1, we replace this indicator function with a smooth $\varepsilon_1$-approximation, as described in Appendix H.3.*

We now present our main result on the derandomization of the JL Lemma.

**Theorem 5.5.** *Let $n$ be unit vectors in $\mathbb{R}^d$, $k$ be the target dimension, $\epsilon$ be a smoothing parameter and $\Delta > 0$ be an accuracy parameter. For any $\varepsilon \geq C\sqrt{\log n/k}$, where $C$ is a sufficiently large constant, initialize $\boldsymbol{M} = \mathbf{0}$ and $\mathbf{\Sigma} = \boldsymbol{I}_{kd}$ and run Algorithm 1 to optimize the $\varepsilon_1$-smoothed version of the objective function in Equation 9 (see Equation 27) using the regularization parameter $\lambda = \frac{\sqrt{\rho/\epsilon^3} + \Delta}{2}$. After $T = \mathcal{O}\left(\text{poly}\left(n, k, d, \log(\delta), \Delta^{-1}\right)\right)$ iterations, this returns a matrix $\boldsymbol{M}$ that satisfies the JL guarantee with distortion at most $\mathcal{O}(\varepsilon)$, with probability $1 - \delta$.*

**Proof.**    Full proof can be found in Appendix H.4. $\qquad\qquad\qquad\qquad\qquad\qquad\qquad\square$

In addition, we have constructed an experiment and show that indeed optimizing a distribution over projection matrices can reduce JL distortion beyond standard Gaussian constructions by directly minimizing worst-case distortion. Full details can be found in Appendix H.5.

## 6    CONCLUSION

We study the theoretical properties of NNs under specific conditions, showing they can discover low-rank structures. Building on (Mousavi-Hosseini et al., 2023), we extend their framework to allow (a) NNs of arbitrary size and depth, (b) all parameters trainable, (c) any smooth loss function, and (d) minimal regularization. The core of our analysis is the *derandomization* Lemma 3.1, which ensures effectiveness even with small regularization. Training biases is a common practice, and our theory guarantees that it can improve model performance. The strength of our lemma is demonstrated in three applications, mainly in NNs and secondarily in MAXCUT and JL embeddings.

Finally, we outline some limitations of our current work and suggest future research directions. Our results rely on the assumption that the input distribution is Gaussian. Extending these findings to other distributions is an interesting avenue for future research. Additionally, it would be valuable to explore connections between our theoretical results and the learning and generalization guarantees that are observed in practice.

## 7 ACKNOWLEDGMENTS AND DISCLOSURE OF FUNDING

The authors would like to thank Alireza Mousavi-Hosseini for useful discussions and feedback. This work has been partially supported by project MIS 5154714 of the National Recovery and Resilience Plan Greece 2.0 funded by the European Union under the NextGenerationEU Program. Ioannis Mitliagkas acknowledges support by Archimedes, Athena Research Center, Greece and a Canada CIFAR AI chair. The majority of work was performed at Archimedes in Athens.

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

## A  RELATED WORK

**Feature learning in NNs.**  Despite the established importance of feature learning in NNs, the specifics of how gradient-based algorithms develop useful features remain somewhat unclear. The neural tangent kernel (NTK) framework, primarily used for examining overparameterized NNs, suggests that neuron movement from their initial positions is minimal, highlighting the role of NN architecture and initial settings (Jacot et al., 2018; Du et al., 2018; Allen-Zhu et al., 2019; Chizat et al., 2019). Limitations of the NTK framework have led researchers to explore other analytical approaches, such as mean-field analysis, initially requiring vast neuron counts (Chizat & Bach, 2018; Mei et al., 2018). Later studies have shown that early stages of training, such as initial steps in GD, are crucial for effective feature learning, with the first layer in two-layer NNs capturing valuable features (Daniely & Malach, 2020; Abbe et al., 2021; 2022; Zhou & Ge, 2024). This early capture of features by the first layer offers better performance than models relying solely on kernel or random features. In the exploration of NN and kernel method interconnections, it has become evident that gradient-based training facilitates representation learning, setting NNs apart from kernel methods (Mousavi-Hosseini et al., 2023; Abbe et al., 2022; Ba et al., 2022; Barak et al., 2022; Damian et al., 2022). A two-layer NN with untrained, randomly initialized weights epitomizes a random features model (Rahimi & Recht, 2007), capturing complex phenomena seen in NN practice (Louart et al., 2018; Mei & Montanari, 2022). Despite inheriting positive traits from optimization procedures, these cannot be fully expressed as random feature regression. The implicit regularization aims of the training dynamics, favoring low-complexity models, are widely discussed (Neyshabur et al., 2015a).

**Single/Multi index models.**  NNs are widely studied for learning single-index and multi-index models, which depend on a few directions in high-dimensional inputs. Recent works demonstrate the effectiveness of two-layer NN in learning single-index (Soltanolkotabi, 2017; Yehudai & Ohad, 2020; Frei et al., 2020; Wu, 2022; Bietti et al., 2022; Xu & Du, 2023; Mahankali et al., 2023; Berthier et al., 2024) and multi-index models (Damian et al., 2022; Bietti et al., 2025; Glasgow, 2024; Suzuki et al., 2024). These studies emphasize the benefits of feature learning over fixed random features. For multi-index functions representable by compact two-layer NN, a GD variant with weight decay can recover ground-truth directions. Gradient-based learning shows that NNs trained via GD can learn useful representations for single-index (Ba et al., 2022; Bietti et al., 2022; Mousavi-Hosseini et al., 2023; Berthier et al., 2024; Oko et al., 2024) and multi-index models (Damian et al., 2022; Abbe et al., 2022; Bietti et al., 2025). Learning complexity is influenced by the information exponent (Arous et al., 2021) or leap complexity (Abbe et al., 2023). While guarantees for low-dimensional models often lead to superpolynomial dependence, other research examines cases where student NN match the target function's architecture (Gamarnik et al., 2025; Akiyama & Suzuki, 2021; Zhou et al., 2022; Veiga et al., 2022; Martin et al., 2024). This study considers an intermediate case where width scales with dimensionality without assuming a known nonlinear activation, showing GD achieves polynomial sample complexity when target weights are diverse. Additionally, statistical query algorithms address related polynomial regression tasks (Dudeja & Hsu, 2018; Chen & Meka, 2020; Garg et al., 2020; Diakonikolas et al., 2024).

A significant line of recent work investigates the learnability of single-index models via Hermite decompositions under Gaussian inputs. These works show that for single-index targets, the first nonzero Hermite term, captured by the information exponent or, in newer formulations, the generative exponent, governs the difficulty of recovering the index direction using first-order or statistical query-style methods (Arous et al., 2021; Wang et al., 2024; Braun et al., 2025). Recent lower bounds based on the generative exponent reveal computational–statistical gaps, establishing sharp statistical query and low-degree polynomial hardness results (Damian et al., 2024; 2025). Complementary

algorithmic results show that gradient-based learners can match these limits in certain regimes: two-layer networks with data reuse effectively reduce the relevant Hermite order (Lee et al., 2024), and new SGD-based methods achieve sample complexities near the generative-exponent boundary (Chen et al., 2025).

# B  PROOF OF MAIN LEMMA

## B.1  PROOF OF LEMMA 3.1

**Lemma.**  Let $x \sim \mathcal{N}(\mathbf{0}, \mathbf{I}_d)$ be a standard Gaussian random variable. For the objective function defined in Equation 2, with $\lambda > \frac{\sqrt{K\rho}}{2}$ where $g_\theta(\cdot)$ satisfies Assumption 2.3, any $\rho$-SOSP satisfies $\|\mathbf{W}\|_F \leq \frac{\rho}{2\lambda - \sqrt{K\rho}}$.

**Proof.**  The first and second derivatives of $f(\mathbf{W}, \mathbf{b}, \theta)$ with respect to $\mathbf{b}$ are given by:

$$\frac{\partial f(\mathbf{W}, \mathbf{b}, \theta)}{\partial b} = \mathbb{E}_{\mathbf{x}}[\nabla g_\theta(\mathbf{W}\mathbf{x} + \mathbf{b})].$$

$$\frac{\partial^2 f(\mathbf{W}, \mathbf{b}, \theta)}{\partial^2 b} = \mathbb{E}_{\mathbf{x}}[\nabla^2 g_\theta(\mathbf{W}\mathbf{x} + \mathbf{b})].$$

The first derivative with respect to $\mathbf{W}$ is:

$$\frac{\partial f(\mathbf{W}, \mathbf{b}, \theta)}{\partial \mathbf{W}} = \mathbb{E}_{\mathbf{x}}[\nabla g_\theta(\mathbf{W}\mathbf{x} + \mathbf{b})\mathbf{x}^\top] + 2\lambda\mathbf{W}.$$

Using Stein's Lemma in the multivariate case, this can be rewritten as:

$$\frac{\partial f(\mathbf{W}, \mathbf{b}, \theta)}{\partial \mathbf{W}} = \mathbb{E}_{\mathbf{x}}[\nabla^2 g_\theta(\mathbf{W}\mathbf{x} + \mathbf{b}) + 2\lambda \mathbf{I}]\mathbf{W}. \tag{10}$$

At a $\rho$-second-order stationary point, the Hessian with respect to $\mathbf{b}$ satisfies:

$$\frac{\partial^2 f(\mathbf{W}, \mathbf{b}, \theta)}{\partial^2 b} = \mathbb{E}_{\mathbf{x}}[\nabla^2 g_\theta(\mathbf{W}\mathbf{x} + \mathbf{b})] \succcurlyeq -\sqrt{K\rho}\mathbf{I}. \tag{11}$$

From Equation 11, it follows that:

$$\mathbb{E}_{\mathbf{x}}[\nabla^2 g_\theta(\mathbf{W}\mathbf{x} + \mathbf{b})] + 2\lambda\mathbf{I} \succeq 2\lambda\mathbf{I} - \sqrt{K\rho}\mathbf{I}.$$

Then dividing both sides with $2\lambda - \sqrt{K\rho}$, we get:

$$\mathbb{E}_{\mathbf{x}}\left[\frac{\nabla^2 g_\theta(\mathbf{W}\mathbf{x} + \mathbf{b}) + 2\lambda\mathbf{I}}{2\lambda - \sqrt{K\rho}}\right] \succeq \mathbf{I}. \tag{12}$$

Using the approximate first-order optimality condition $\left\|\frac{\partial f(W,b)}{\partial W}\right\|_F < \rho$ along with Equations 10 and 12, we have:

$$\frac{\rho}{2\lambda - \sqrt{K\rho}} \geq \left\|\mathbb{E}_{\mathbf{x}}\left[\frac{\nabla^2 g_\theta(\mathbf{W}\mathbf{x} + \mathbf{b}) + 2\lambda\mathbf{I}}{2\lambda - \sqrt{K\rho}}\right]\mathbf{W}\right\|_F$$

$$\geq \sigma_{\min}\left(\mathbb{E}_{\mathbf{x}}\left[\frac{\nabla^2 g_\theta(\mathbf{W}\mathbf{x} + \mathbf{b}) + 2\lambda\mathbf{I}}{2\lambda - \sqrt{K\rho}}\right]\right)\|\mathbf{W}\|_F$$

$$\geq \|\mathbf{W}\|_F,$$

where $\sigma_{\min}$ in the penultimate inequality is the minimum singular value which is lower bounded by one.

$\square$

## B.2 PROOF OF LEMMA 3.1 FOR $\rho = 0$

**Lemma.** Let $\boldsymbol{x} \sim \mathcal{N}(\boldsymbol{0}, \boldsymbol{I}_d)$ be a standard Gaussian random variable. For the objective function defined in Equation 2, with $\lambda > \frac{\sqrt{K}\rho}{2}$ where $g_\theta(\cdot)$ satisfies Assumption 2.3, any second-order stationary point satisfies $\boldsymbol{W} = \boldsymbol{0}$.

**Proof.** The first and second derivatives of $f(\boldsymbol{W}, \boldsymbol{b}, \theta)$ with respect to $\boldsymbol{b}$ are given by:

$$\frac{\partial f(\boldsymbol{W}, \boldsymbol{b}, \theta)}{\partial \boldsymbol{b}} = \mathbb{E}_{\boldsymbol{x}}[\nabla g_\theta(\boldsymbol{W}\boldsymbol{x} + \boldsymbol{b})].$$

$$\frac{\partial^2 f(\boldsymbol{W}, \boldsymbol{b}, \theta)}{\partial^2 \boldsymbol{b}} = \mathbb{E}_{\boldsymbol{x}}[\nabla^2 g_\theta(\boldsymbol{W}\boldsymbol{x} + \boldsymbol{b})].$$

The first derivative with respect to $\boldsymbol{W}$ is:

$$\frac{\partial f(\boldsymbol{W}, \boldsymbol{b}, \theta)}{\partial \boldsymbol{W}} = \mathbb{E}_{\boldsymbol{x}}[\nabla g_\theta(\boldsymbol{W}\boldsymbol{x} + \boldsymbol{b})\boldsymbol{x}^\top] + 2\lambda\boldsymbol{W}. \tag{13}$$

Using Stein's Lemma in the multivariate case, this can be rewritten as:

$$\frac{\partial f(\boldsymbol{W}, \boldsymbol{b}, \theta)}{\partial \boldsymbol{W}} = \mathbb{E}_{\boldsymbol{x}}[\nabla^2 g_\theta(\boldsymbol{W}\boldsymbol{x} + \boldsymbol{b}) + 2\lambda\boldsymbol{I}]\boldsymbol{W}. \tag{14}$$

At a second-order stationary point, the Hessian with respect to $b$ satisfies:

$$\frac{\partial^2 f(\boldsymbol{W}, \boldsymbol{b}, \theta)}{\partial^2 \boldsymbol{b}} = \mathbb{E}_{\boldsymbol{x}}[\nabla^2 g_\theta(\boldsymbol{W}\boldsymbol{x} + \boldsymbol{b})] \succcurlyeq 0. \tag{15}$$

From Equation 15, it follows that:

$$\mathbb{E}_{\boldsymbol{x}}[\nabla^2 g_\theta(\boldsymbol{W}\boldsymbol{x} + \boldsymbol{b})] + 2\lambda\boldsymbol{I} \succ 0, \tag{16}$$

where the addition of $2\lambda\boldsymbol{I}$ ensures strict positive definiteness.

Using the first-order optimality condition $\frac{\partial f(\boldsymbol{W}, \boldsymbol{b}, \theta)}{\partial \boldsymbol{W}} = 0$ along with Equations 14 and 16, we have:

$$\mathbb{E}_{\boldsymbol{x}}[\nabla^2 g_\theta(\boldsymbol{W}\boldsymbol{x} + \boldsymbol{b}) + 2\lambda\boldsymbol{I}]\boldsymbol{W} = \boldsymbol{0}. \tag{17}$$

Since $\mathbb{E}_{\boldsymbol{x}}[\nabla^2 g(\boldsymbol{W}\boldsymbol{x} + \boldsymbol{b})] + 2\lambda\boldsymbol{I} \succ 0$ (from Equation 2), the only solution is $\boldsymbol{W} = \boldsymbol{0}$.

$\square$

## C OPTIMIZATION ALGORITHMS

Below we give the two main algorithms for finding SOSPs; PGD and Hessian Descent.

---

**Algorithm 1** Perturbed Gradient Descent

---

**Require:** Objective function $f(\boldsymbol{x})$, initial point $\boldsymbol{x}_0$, gradient Lipschitz constant $L$, learning rate $\eta = \frac{1}{L}$, maximum iterations $T$
1: Initialize $\boldsymbol{x}_1 \leftarrow \boldsymbol{x}_0$
2: **for** $t = 1$ to $T$ **do**
3:     **if** perturbation condition holds **then**
4:         Draw random perturbation $\boldsymbol{\xi}_t$
5:         $\boldsymbol{x}_t \leftarrow \boldsymbol{x}_t + \boldsymbol{\xi}_t$
6:     **end if**
7:     $\boldsymbol{x}_{t+1} \leftarrow \boldsymbol{x}_t - \eta\nabla f(\boldsymbol{x}_t)$
8: **end for**
9: **return** $\boldsymbol{x}_{T+1}$

---

---

**Algorithm 2** Hessian Descent

---

**Require:** Gradient $\nabla g$, Hessian $\nabla^2 g$, initial point $\boldsymbol{x}_0$, step size $\nu = \frac{1}{L}$, perturbation step size $h = \frac{3\sqrt{\rho}}{K}$, Lipschitz constants $L$, $K$, $\rho$
  1: Initialize $t \leftarrow 0$
  2: **while** true **do**
  3:      **if** $\|\nabla g(\boldsymbol{x}_t)\| > \rho$ **then**
  4:          $\boldsymbol{x}_{t+1} \leftarrow \boldsymbol{x}_t - \nu \cdot \nabla g(\boldsymbol{x}_t)$
  5:      **else if** $\|\nabla g(\boldsymbol{x}_t)\| \leq \rho$ **and** $\lambda_{\min}(\nabla^2 g(\boldsymbol{x}_t)) < -\sqrt{K\rho}$ **then**
  6:          $\boldsymbol{u}_1 \leftarrow$ eigenvector corresponding to $\lambda_{\min}(\nabla^2 g(\boldsymbol{x}_t))$
  7:          $\boldsymbol{x}_{t+1} \leftarrow \boldsymbol{x}_t + h\boldsymbol{u}_1$
  8:      **else**
  9:          **return** $\boldsymbol{x}_t$
 10:      **end if**
 11:     $t \leftarrow t + 1$
 12: **end while**

---

# D    REFORMULATION

$$
\begin{aligned}
R(\boldsymbol{W}_\|, \boldsymbol{W}_\perp, \boldsymbol{b}; \theta) &= \mathbb{E}_{\boldsymbol{x}_\perp, \boldsymbol{x}_\|} \left[ \ell\big(h'(\boldsymbol{x}_\|), g_\theta(\boldsymbol{W}_\perp \boldsymbol{x}_\perp + \boldsymbol{W}_\| \boldsymbol{x}_\| + \boldsymbol{b})\big) \right] + \lambda \|\boldsymbol{W}_\| + \boldsymbol{W}_\perp\|_F^2 \\
&= \mathbb{E}_{\boldsymbol{x}_\perp} \left[ \mathbb{E}_{\boldsymbol{x}_\|} \left[ \ell\big(h'(\boldsymbol{x}_\|), g_\theta(\boldsymbol{W}_\perp \boldsymbol{x}_\perp + \boldsymbol{W}_\| \boldsymbol{x}_\| + \boldsymbol{b})\big) \right] \right] \\
&\quad + \lambda \|\boldsymbol{W}_\|\|_F^2 + \lambda \|\boldsymbol{W}_\perp\|_F^2 \\
&= \mathbb{E}_{\boldsymbol{x}_\perp} \left[ \ell'_{\theta'}(\boldsymbol{W}_\perp \boldsymbol{x}_\perp + \boldsymbol{b}) \right] + \lambda \|\boldsymbol{W}_\perp\|_F^2
\end{aligned} \tag{18}
$$

where $\ell'_{\theta'}(\boldsymbol{W}_\perp \boldsymbol{x}_\perp + \boldsymbol{b}) := \mathbb{E}_{\boldsymbol{x}_\|} \left[ \ell\left(h'\left(\boldsymbol{x}_\|\right), g_\theta\left(\boldsymbol{W}_\perp \boldsymbol{x}_\perp + \boldsymbol{W}_\| \boldsymbol{x}_\| + \boldsymbol{b}\right)\right) \right] + \lambda \|\boldsymbol{W}_\|\|_F^2$. Equation 18 holds because $\boldsymbol{W}_\perp$ is orthogonal to $\boldsymbol{W}_\|$. For notational convenience, we suppress $\boldsymbol{W}_\|$, $\boldsymbol{W}_\perp$ and $\boldsymbol{x}_\perp$ in the expectation and just write

$$
R(\boldsymbol{W}_\perp, \boldsymbol{b}; \theta') = \mathbb{E}_{\boldsymbol{x}_\perp} \left[ \ell'_{\theta'}(\boldsymbol{W}_\perp \boldsymbol{x}_\perp + \boldsymbol{b}) \right] + \lambda \|\boldsymbol{W}_\perp\|_F^2.
$$

Additionally, we clarify that our analysis operates under the assumption that the optimization method employed converges to a $\rho$-SOSP with respect to all model parameters, including those encapsulated in $\theta$. Specifically, we assume that an approximate SOSP is identified jointly over the entire parameter space. Once such a point is found, we observe that fixing the auxiliary parameters $\theta$ preserves the approximate second-order stationarity with respect to the first-layer weight matrix $\boldsymbol{W}$.

# E    PROOFS OF SECTION 4

## E.1    PROOF OF THEOREM 4.2

**Theorem.** Let the objective function in Equation 5 be twice differentiable, bounded below, and satisfies Assumption 2.3. Let the data $\boldsymbol{x} \sim \mathcal{N}(\boldsymbol{0}, \boldsymbol{I}_d)$, and suppose labels are generated according to Equation 3. Let $\rho > 0$ be a prespecified accuracy, and define the regularization parameter $\lambda = \frac{\sqrt{K\rho} + \Delta}{2}$, where $K$ is the Hessian Lipschitz constant and $\Delta > 0$ is an arbitrarily small constant. Then, with probability $1 - \delta$, running Algorithm 1 for $T > \mathcal{O}\left(\text{poly}\left(L, \log(d), \log(\delta), \varepsilon^{-1}, \Delta^{-1}\right)\right)$ iterations with a step size $\mathcal{O}(1/L)$, yields a weight matrix $\boldsymbol{W} = \boldsymbol{W}_\perp + \boldsymbol{W}_\|$ that satisfies:

$$
\|\boldsymbol{W}_\perp\|_F < \varepsilon.
$$

**Proof.** Since the objective function has Lipschitz continuous gradient and Hessian and is bounded from below, this implies that it has at least one $\rho$-SOSP. Choose some $\Delta > 0$ and $\lambda = \frac{\sqrt{K\rho} + \Delta}{2}$ and using Lemma 3.1, we get that any $\rho$-SOSP is a weight matrix $\boldsymbol{W} = \boldsymbol{W}_\perp + \boldsymbol{W}_\|$, that satisfies $\|\boldsymbol{W}_\perp\|_F \leq \frac{\rho}{\Delta}$.

Let $\varepsilon = \frac{\rho}{\Delta}$, solving for $\rho$, this gives $\rho = \varepsilon\Delta$. Then, running Algorithm 1 for:

$$T = \mathcal{O}\left(\frac{L}{\rho^2}\log^4(d) - \frac{L}{\rho^2}\log^4(\delta)\right) = \mathcal{O}\left(\frac{L}{\varepsilon^2\Delta^2}\log^4(d) - \frac{L}{\varepsilon^2\Delta^2}\log^4(\delta)\right),$$

iterations gives the required result. $\qquad\square$

## F    EXPERIMENTS IN NEURAL NETWORKS OF SECTION 4.3

The student NN is a two-layer feedforward network with a single hidden layer. The input data $\boldsymbol{X} \in \mathbb{R}^{n \times d}$ is sampled from a standard multivariate Gaussian distribution with $d = 2$ and $n = 50$. The network architecture consists of an input layer with $d$ features, a hidden layer of width $h = 1,000$, and an output layer that produces scalar predictions.

The first-layer weight matrix $\boldsymbol{W} \in \mathbb{R}^{h \times d}$ is initialized with entries drawn from $\mathcal{N}(0, 1/d)$, while the bias vector $\boldsymbol{b} \in \mathbb{R}^h$ and the second-layer weight vector $\boldsymbol{a}$ are initialized from $\mathcal{N}(0, 1/h^2)$. The trainable parameters include both $\boldsymbol{W}$ and $\boldsymbol{b}$ and we freeze the second layer for stability reasons. The student NN computes predictions according to the mapping:

$$y = \boldsymbol{a}^\top \mathrm{ReLU}_2(\boldsymbol{W}\boldsymbol{x} + \boldsymbol{b}),$$

where, $\mathrm{ReLU}_2$ denotes a smoothed version of the ReLU, as defined in Section 4.2. Although $\mathrm{ReLU}_2$ is used in our experiments, other nonlinearities such as tanh can be employed in the same framework.

To generate the labels, we use a teacher network based on a single-index model. A fixed direction $\theta$ is chosen as

$$\theta = \frac{1}{\sqrt{2}}(1, 1)^\top,$$

and labels are generated according to the rule:

$$y = \tanh(\theta \cdot \boldsymbol{x}) + \varepsilon, \text{ where } \varepsilon \sim \mathcal{N}(0, 0.1).$$

Figure 3 shows that the first-layer weights of the student network have effectively converged to the principal subspace, indicating that the network has focused on the relevant direction.

## G    SUPPORTING MATERIAL OF SECTION 5.1

### G.1    PROOF OF THEOREM 5.2

**Theorem.** Let $G = (V, E)$ be a graph with $m$ edges, where the edge weights are given by $w_{i,j} = w_{j,i} = 1$ for all $(i, j) \in E$. Let $\boldsymbol{V} \in \mathbb{R}^{m \times m}$ be the matrix of vectors obtained from the SDP relaxation of the MAXCUT problem, as described in (Goemans & Williamson, 1995). Denote by $\boldsymbol{z} \sim \mathcal{N}(\boldsymbol{0}, \boldsymbol{I}_m)$ a standard multivariate Gaussian vector, and let $\boldsymbol{\mu}$ represent a vector of means. Initialize Algorithm 1 with $\boldsymbol{\mu} = \boldsymbol{0}$, and optimize the $\epsilon$-smoothed version of the objective function in Equation 7 (see Equation 27), using the regularization parameter $\lambda = \frac{\sqrt{\rho/\epsilon^3 + \Delta}}{2}$, and run for $T = \mathcal{O}\left(\mathrm{poly}\left(\log(m), \log(\delta), \epsilon^{-1}, \Delta^{-1}\right)\right)$ iterations. After this optimization process, the resulting vector $\boldsymbol{\mu}$ defines a cut whose value is guaranteed to be at least:

$$\mathrm{OPT}(\alpha - \mathcal{O}(\epsilon)),$$

with probability $1 - \delta$. Here, $\alpha = 0.878$ is the approximation factor from (Goemans & Williamson, 1995).

**Proof.** Given a graph with $m$ edges and weights that are equal to one, $w_{i,j} = w_{j,i} = 1$, we aim to provide an approximation to the MAXCUT problem by derandomizing the randomized rounding algorithm. Consider the matrix $\boldsymbol{V} \in \mathbb{R}^{m \times m}$ which gathers all the vectors from the Semidefinite Program. Let us also define the function $\boldsymbol{V} \boldsymbol{z} + \boldsymbol{\mu}$, where $\boldsymbol{z} \sim \mathcal{N}(\boldsymbol{0}, \boldsymbol{I}_m)$ and $\boldsymbol{\mu}$ is a vector of means. Our objective is to minimize the negative expected cut:

$$f(\boldsymbol{V}, \boldsymbol{\mu}) = -\sum_{i<j} w_{i,j} \Pr \left[ \mathrm{sgn}(\boldsymbol{v}_i \cdot \boldsymbol{z} + \mu_i) \neq \mathrm{sgn}(\boldsymbol{v}_j \cdot \boldsymbol{z} + \mu_j) \right],$$

for which initially we have $\boldsymbol{\mu} = \boldsymbol{0}$. To minimize this function using our Lemma, we define the indicator function:

$$I(x, y) = \begin{cases} 1, & \text{if } xy < 0 \\ 0, & \text{if } xy \geq 0 \end{cases}$$

Using this indicator, we can reformulate the objective as:

$$f\left(\boldsymbol{V} \boldsymbol{z} + \boldsymbol{\mu}\right) = -\mathbb{E}_{\boldsymbol{z}} \left[ \sum_{i<j} w_{i,j} I(\boldsymbol{v}_i \cdot \boldsymbol{z} + \mu_i, \boldsymbol{v}_j \cdot \boldsymbol{z} + \mu_j) \right] + \lambda \|\boldsymbol{V}\|_F^2, \tag{19}$$

We are concerned with the value of $I(\boldsymbol{v}_i \cdot \boldsymbol{z}, \boldsymbol{v}_j \cdot \boldsymbol{z})$. If $I(\boldsymbol{v}_i \cdot \boldsymbol{z}, \boldsymbol{v}_j \cdot \boldsymbol{z}) = 1$, then this edge contributes to the cut because the signs are different, otherwise it does not. However, since this function is not smooth, we introduce a smoothed version:

$$\tilde{I}(x, y) = \begin{cases} 1, & \text{if } xy < 0 \text{ and } |x|, |y| > \epsilon \\ 0, & \text{if } |x| < \frac{\epsilon}{2} \text{ or } |y| < \frac{\epsilon}{2} \text{ or } xy > 0 \\ [0, 1], & \text{otherwise.} \end{cases}$$

This function is smoothed to be twice differentiable in the interval $[0, 1]$, and we can approximate it as a polynomial to ensure that the Hessian Lipschitz constant is $K = \mathcal{O}\left(\frac{1}{\epsilon^3}\right)$ and the gradient Lipschitz constant is $L = \mathcal{O}\left(\frac{1}{\epsilon^2}\right)$.

The function $\tilde{I}(x, y)$ is formally defined as follows. First, we define,

$$S(x) = \begin{cases} 1, & x \geq \epsilon \\ 0, & x < \frac{\epsilon}{2} \\ \frac{8}{\epsilon^2}\left(x - \frac{\epsilon}{2}\right)^2, & \frac{\epsilon}{2} < x \leq \frac{3\epsilon}{4} \\ -\frac{8}{\epsilon^2}(x - \epsilon)^2 + 1, & \frac{3\epsilon}{4} < x < \epsilon \end{cases}$$

then, the smoothed step function $\tilde{I}$ is defined as:

$$\tilde{I}(x, y) = S(x)S(-y) + S(-x)S(y).$$

By using this smoothed function to calculate the cut, we worsen the result from the original cut by at most $\mathcal{O}(m\epsilon)$, where $m$ is the number of edges. This corresponds to the area of disagreement between using the actual indicator function and the smoothed indicator function. Our goal is now to minimize the following function:

$$f\left(\boldsymbol{V} \boldsymbol{z} + \boldsymbol{\mu}\right) = \mathbb{E}_{\boldsymbol{z}} \left[ \sum_{i<j} \tilde{w}_{i,j} \tilde{I}(\boldsymbol{v}_i \cdot \boldsymbol{z} + \mu_i, \boldsymbol{v}_j \cdot \boldsymbol{z} + \mu_j) \right] + \lambda \|\boldsymbol{V}\|_F^2, \tag{20}$$

where we absorb the minus sign into the original definition of $w_{i,j}$, for convenience. Choose $\rho$ such that at the end of the optimization we have $\|\boldsymbol{V}\|_F^2 < \epsilon^3$.

Next, we investigate the effect of ignoring $V$ and only using the means $\mu$. At the end of the optimization, we have vectors $v_i$, $i = 1, \ldots, m$, with $\|v_i\|_2^3 \leq \epsilon^2$, for all $i$. We examine two cases for each edge of the graph:

1) If $\mu_i \mu_j < 0$, these two values contribute to the cut directly, so we do not need anything more.

2) If $\mu_i \mu_j > 0$, we need further analysis. Without loss of generality, assume that $\mu_i > 0$ and $\mu_j > 0$.

To analyze the difference when using full randomness, we need to consider the following. For the edge to contribute to the randomized version, we require that either $v_i \cdot z + \mu_i < -\epsilon$ or $v_j \cdot z + \mu_j < -\epsilon$, since we want one of the terms to change signs and thus contribute to $\tilde{I}$. Consequently, we would have to generate a $z$ such that the magnitude $\|v_i \cdot z\| > \epsilon$, which has exponentially small probability, that is:

$$\Pr(\|v_i \cdot z\| > \epsilon) \leq 2 \exp\left(-\frac{1}{\epsilon^4}\right).$$

Because of this we can conclude that the total expected cut remains nearly unchanged.

So, initially before smoothing, we have $\sum_{i<j} \mathbb{E}_z \tilde{w}_{i,j}\left[I(v_i \cdot z, v_j \cdot z)\right] = \alpha\text{OPT}$, then after using the smoothed function we obtain:

$$\sum_{i<j} \mathbb{E}_z \tilde{w}_{i,j}\left[\tilde{I}(v_i \cdot z, v_j \cdot z)\right] = \alpha\text{OPT} - \mathcal{O}(\epsilon m).$$

Finally, after optimization, we have:

$$\sum_{i<j} \tilde{w}_{i,j} \mathbb{E}_z\left[\tilde{I}(v_i \cdot z + \mu_i, v_j \cdot z + \mu_j)\right] \geq \alpha\text{OPT} - \mathcal{O}(\epsilon m),$$

As a final step, we compare our (deterministic) cut, with what would have happened if we took the original randomized version. To complete this step, we use a union bound:

$$\sum_{i<j} \tilde{w}_{i,j} \tilde{I}(\mu_i, \mu_j) = \alpha\text{OPT} - \mathcal{O}(\epsilon m) - \Pr\left(|v_i z| > \epsilon \text{ for any edge}\right)$$

$$\geq \alpha\text{OPT} - \mathcal{O}(\epsilon m) - m \Pr\left(|v_i z| > \epsilon\right)$$

$$\geq \alpha\,\text{OPT} - \mathcal{O}(\epsilon m) - \mathcal{O}\left(m \cdot \exp\left(-\frac{1}{\epsilon^4}\right)\right)$$

$$\geq \alpha\,\text{OPT} - \mathcal{O}(\epsilon m) - \mathcal{O}(\epsilon m)$$

$$= \alpha\,\text{OPT} - \mathcal{O}(\epsilon\,\text{OPT})$$

$$= \text{OPT} \cdot \left(\alpha - \mathcal{O}(\epsilon)\right),$$

where the penultimate equality follows since OPT is a function of the edges $m$.

For the iteration complexity, since we want to reach a point with $\|V\|_F < \epsilon^{3/2}$, for the specific choice of $\lambda = \frac{\sqrt{\rho/\epsilon^3 + \Delta}}{2}$, for $\rho = \Delta\epsilon^{3/2}$ and running Algorithm 1 with step size $\mathcal{O}(\epsilon^2)$:

$$T = \mathcal{O}\left(\frac{L}{\rho^2}\log^4(m) - \frac{L}{\rho^2}\log^4(\delta)\right) = \mathcal{O}\left(\frac{1}{\epsilon^5\Delta^2}\log^4(m) - \frac{1}{\epsilon^5\Delta^2}\log^4(\delta)\right),$$

iterations gives a $\rho-$SOSP with probability $1-\delta$ and thus $\|V\| \leq \epsilon^{3/2}$ (via Theorem 4.2). Combining the above and the fact that we have reached a $\rho-$SOSP giving the required result. $\qquad\square$

## G.2 EXPERIMENTS IN MAXCUT

In this experiment, we evaluated a stochastic optimization algorithm for solving the MAXCUT problem on a randomly generated undirected graph with $m = 15$ vertices and an edge probability

of 0.6. The exact MAXCUT value (computed as 41) was obtained using exhaustive search and used as a ground-truth reference. The optimization procedure is based on the Goemans-Williamson relaxation, where node embeddings are derived from the top eigenvectors of the adjacency matrix. A stochastic gradient-based method is then applied, which samples noisy directions from a Gaussian distribution parameterized by a mean vector $m$ and log-standard deviation $\log \sigma$. and the number of cut edges is evaluated. Gradients with respect to both $m$ and $\sigma$ are estimated using a Monte Carlo approximation with 100 samples per step. The parameters are updated via SGD, which is sufficient for this task, with adaptive learning rates: 0.01 for $m$ and 0.001 for $\sigma$. To ensure stability and exploration, a regularization term is applied to $\sigma$, and its values are clipped to the range $[10^{-3}, 1.5]$. Learning rates are further annealed by decay factors every 100 iterations. The algorithm was run for 5000 iterations. Throughout optimization, we tracked the evolution of cut values, the maximum standard deviation across dimensions, and sampled cut edges to monitor progress relative to the exact MAXCUT benchmark.

Figure 4 illustrates the progression of the cut value over the course of training, showing consistent improvement and eventual convergence to the optimal cut obtained via brute force. In parallel, Figure 5 shows the evolution of the maximum value of $\sigma^2$, which steadily decreased over time. This trend indicates that the algorithm gradually reduced its randomness as it converged toward a confident, high-quality solution. Notably, our optimization method substantially outperformed the baseline cut value of approximately 36 achieved by the classical randomized algorithm. Overall, the results demonstrate that the method not only successfully identified an optimal cut but also naturally annealed its uncertainty, confirming both its effectiveness and stability.

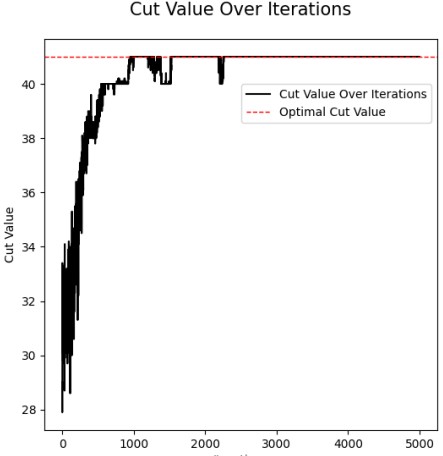
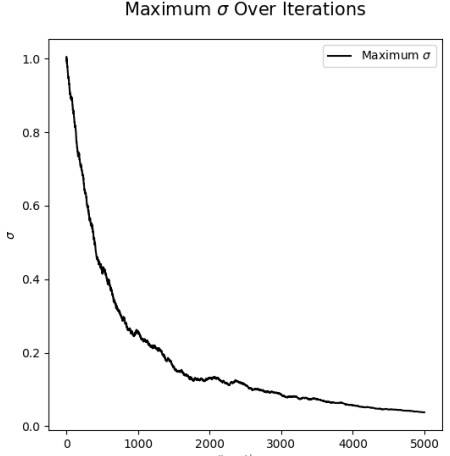

Figure 4: Progress of the cut value over iterations.

Figure 5: Evolution of the maximum $\sigma^2$ value over iterations.

# H    SUPPORTING MATERIAL OF SECTION 5.2

## H.1    REFORMULATION OF JOHNSON-LINDENSTRAUSS OBJECTIVE FUNCTION

To achieve the JL guarantee from Definition 5.3 we define a linear mapping $f(\boldsymbol{x}) = \mathbf{A}\boldsymbol{x}$, where $\mathbf{A} \in \mathbb{R}^{k \times d}$. The JL Lemma guarantees the existence of a random linear mapping that achieves this projection with high probability:

**Lemma H.1** (Distributional JL Lemma). *For $\varepsilon, \delta \in (0, 1)$ and $k = \mathcal{O}(\log(1/\delta)/\varepsilon^2)$, there exists a probability distribution $D$ over linear functions $f : \mathbb{R}^d \to \mathbb{R}^k$ such that for every $x \in \mathbb{R}^d$:*

$$\Pr_{f \sim D} \left( \|f(\boldsymbol{x})\|_2^2 \in \left[ (1-\varepsilon)\|\boldsymbol{x}\|_2^2, (1+\varepsilon)\|\boldsymbol{x}\|_2^2 \right] \right) \geq 1 - \delta.$$

Let $\mathbf{A}$ be a random matrix whose elements $a_{i,j}$ are independently drawn from a Gaussian distribution with mean $\mu_{i,j}$ and variance $\sigma_{i,j}^2$. Define the distortion function as:

$$h(\mathbf{A}; \boldsymbol{x}_i) = \left| \|\mathbf{A}\boldsymbol{x}_i\|_2^2 - 1 \right|, \tag{21}$$

where $\mathbf{A} \sim \mathcal{N}(\boldsymbol{M}, \boldsymbol{\Sigma})$. Our objective is to minimize the following function:

$$f(\mathbf{A}; \boldsymbol{x}_i) = \sum_{i=1}^{n} \mathrm{Pr}\left(h(\mathbf{A}; \boldsymbol{x}_i) > \varepsilon\right) + \frac{\|\boldsymbol{\Sigma}^{1/2}\|_F^2}{2kd}, \tag{22}$$

where $\boldsymbol{\Sigma}^{1/2}$ denotes a matrix whose entries are the square roots of the corresponding variances in $\boldsymbol{\Sigma}$.

The first term measures the number of distortion violations (i.e., how often the projected norm deviates from 1 by more than $\varepsilon$), while the second term is a regularization penalty on the variance of the matrix entries.

Applying a union bound, the objective in Equation 22 serves as an upper bound for:

$$\mathrm{Pr}\left(\max_{i=1,\ldots,n} h(\mathbf{A}; \boldsymbol{x}_i) > \varepsilon\right) + \frac{\|\boldsymbol{\Sigma}^{1/2}\|_F^2}{2kd}, \tag{23}$$

which represents the probability that the maximum distortion across all data points exceeds $\varepsilon$, plus a regularization term. As the variances in $\boldsymbol{\Sigma}$ approach zero, this regularizer vanishes. In the context of the JL Lemma, our goal is to minimize the probability expressed in Equation 23.

To use Lemma 3.1 we can think of the matrix $\mathbf{A}$ as a $kd$-dimensional vector and write:

$$\mathbf{A}^{vec} = \boldsymbol{\Sigma}^{1/2}\boldsymbol{z} + \boldsymbol{\mu}, \tag{24}$$

where $\boldsymbol{\mu} = (\mu_{1,1}, \mu_{2,1}, \ldots, \mu_{k,d})^\top \in \mathbb{R}^{kd}$ is a vectorized version of each mean of matrix $\mathbf{A}$, $\boldsymbol{\Sigma} = \mathrm{diag}\left(\sigma_{1,1}^2, \sigma_{1,2}^2, \ldots, \sigma_{k,d}^2\right)^\top \in \mathbb{R}_+^{kd \times kd}$ is the diagonal covariance matrix and $\boldsymbol{z}$ is a $kd$-dimensional multivariate Gaussian vector with independent entries with zero mean and unit variance.

Then, define,

$$g_\theta(\boldsymbol{A}^{vec}) = g_\theta\left(\boldsymbol{\Sigma}^{1/2}\boldsymbol{z} + \boldsymbol{\mu}\right) = \sum_{i=1}^{n} \mathbf{1}_{\left\{h\left(\boldsymbol{\Sigma}^{1/2}\boldsymbol{z} + \boldsymbol{\mu}; x_i\right) > \varepsilon\right\}} \tag{25}$$

Then, we can define the objective function:

$$
\begin{aligned}
f\left(\boldsymbol{\Sigma}^{1/2}, \boldsymbol{\mu}\right) &= \mathbb{E}\left[g_\theta\left(\boldsymbol{\Sigma}^{1/2}\boldsymbol{z} + \boldsymbol{\mu}\right)\right] + \frac{\left\|\boldsymbol{\Sigma}^{1/2}\right\|_F^2}{2kd} \\
&= \sum_{i=1}^{n} \mathbb{E}\left[\mathbf{1}_{\left\{h\left(\boldsymbol{\Sigma}^{1/2}\boldsymbol{z} + \boldsymbol{\mu}; x_i\right) > \varepsilon\right\}}\right] + \frac{\left\|\boldsymbol{\Sigma}^{1/2}\right\|_F^2}{2kd} \\
&= \sum_{i=1}^{n} \mathrm{Pr}\left(h\left(\boldsymbol{\Sigma}^{1/2}\boldsymbol{z} + \boldsymbol{\mu}; x_i\right) > \varepsilon\right) + \frac{\left\|\boldsymbol{\Sigma}^{1/2}\right\|_F^2}{2kd}.
\end{aligned} \tag{26}
$$

**Remark H.2.** *It is important to observe that Equations 22 and 26 represent the same quantity, but are expressed using different parameterizations.*

Thus, minimizing Equation 22 is equivalent to minimizing Equation 26, where optimization is carried out over the parameters $(\boldsymbol{\Sigma}^{1/2}, \boldsymbol{\mu})$.

## H.2 PROOF OF JOHNSON-LINDENSTRAUSS GUARANTEE PRESERVATION LEMMA

Here we give an extension of Lemma 4 from (Tsikouras et al., 2024) which is required due to the practical limitation that achieving an exact SOSP is not feasible. Since Algorithm 1 identifies an approximate $\rho$-SOSP, an additional result is required to provide a stopping criterion once the variance becomes sufficiently small. This ensures that the mean can be used with a controlled deterioration of the JL guarantee.

**Lemma H.3.** *Given $n$ unit vectors in $\mathbb{R}^d$ and a target dimension $k$, choose $\varepsilon$ such that the random matrix $\mathbf{A} \sim N(\boldsymbol{M}, \boldsymbol{\Sigma})$ satisfies the JL guarantee with distortion $\varepsilon$ with probability at least $1/6$. Then using matrix $\boldsymbol{M}$ instead of sampling from $\mathbf{A}$ retains the JL guarantee with a threshold increased by at most $\mathrm{poly}(\sigma_{\max}, 1/k)$.*

**Proof.** We start with the assumption that $\frac{1}{k}\|\mathbf{A}\boldsymbol{x}\|_2^2 \in (1-\varepsilon, 1+\varepsilon)$ with probability at least $\frac{1}{6}$.

Expressing $\mathbf{A}$ as $\mathbf{A} = \boldsymbol{M} + \mathbf{Z}$ where $\mathbf{Z} \sim N(\boldsymbol{0}, \boldsymbol{\Sigma})$. For this, we have

$$\|\underset{\sim}{\mathbf{Z}}\boldsymbol{x}_2^2\| \leq \|\mathbf{Z}\boldsymbol{x}\|_2^2 \leq \|\widetilde{\mathbf{Z}}\boldsymbol{x}\|_2^2,$$

where $\widetilde{\mathbf{Z}}$ and $\underset{\sim}{\mathbf{Z}}$ are the same as $\mathbf{Z}$ but scaled with the maximum and minimum variance from $\boldsymbol{\Sigma}$ respectively. This way, all the entries of $\widetilde{\mathbf{Z}}$ have the same common variance, $\sigma_{\max}^2$ and all the entries of $\underset{\sim}{\mathbf{Z}}$ have the same common variance, $\sigma_{\min}^2$.

From the JL Lemma, we can select $\varepsilon_0$ such that

$$\frac{1}{k}\|\widetilde{\mathbf{Z}}\boldsymbol{x}\|_2^2 \in [\sigma_{\max}^2(1-\varepsilon_0), \sigma_{\max}^2(1+\varepsilon_0)]$$

$$\frac{1}{k}\|\underset{\sim}{\mathbf{Z}}\boldsymbol{x}\|_2^2 \in [\sigma_{\min}^2(1-\varepsilon_0), \sigma_{\min}^2(1+\varepsilon_0)]$$

with probability at least $\frac{6}{7}$. This ensures there exists an overlap where both inequalities for $\mathbf{A}$, $\widetilde{\mathbf{Z}}$ and $\underset{\sim}{\mathbf{Z}}$ hold simultaneously. Our goal is to determine how much excess distortion we get when using $\boldsymbol{M}$ instead of sampling from the random matrix $A$.

Using the triangle inequality we have:

$$\frac{1}{k}\|\boldsymbol{M}\boldsymbol{x}\|_2 = \frac{1}{k}\|\boldsymbol{M}\boldsymbol{x} + \mathbf{Z}\boldsymbol{x} - \mathbf{Z}\boldsymbol{x}\|_2 \leq \frac{1}{k}\|\boldsymbol{M}\boldsymbol{x} + \mathbf{Z}\boldsymbol{x}\|_2 + \frac{1}{k}\|\mathbf{Z}\boldsymbol{x}\|_2 \leq \frac{1}{k}\|\mathbf{A}\boldsymbol{x}\|_2 + \frac{1}{k}\|\widetilde{\mathbf{Z}}\boldsymbol{x}\|_2,$$

which by squaring both sides and using the JL guarantee for $A$ and $\widetilde{\mathbf{Z}}$, we obtain:

$$\frac{1}{k}\|\boldsymbol{M}\boldsymbol{x}\|_2^2 \leq \frac{1}{k}\|\mathbf{A}\boldsymbol{x}\|_2^2 + \frac{2}{k^2}\|\mathbf{A}\boldsymbol{x}\|_2\|\widetilde{\mathbf{Z}}\boldsymbol{x}\|_2 + \frac{1}{k}\|\widetilde{\mathbf{Z}}\boldsymbol{x}\|_2^2$$

$$\leq 1 + \varepsilon + \frac{2\sigma_{\max}}{k}\sqrt{1+\varepsilon}\sqrt{1+\varepsilon_0} + \sigma_{\max}^2(1+\varepsilon_0)$$

$$\leq 1 + \varepsilon + \frac{2\sqrt{2}\sigma_{\max}}{k}\sqrt{1+\varepsilon} + 2\sigma_{\max}^2.$$

For the lower bound, using the Cauchy-Schwarz inequality and the JL guarantee for $A$ and $\underset{\sim}{\mathbf{Z}}$, we have:

$$\frac{1}{k}\|\boldsymbol{M}\boldsymbol{x}\|_2^2 \geq \frac{1}{2k}\|\boldsymbol{M}\boldsymbol{x} + \mathbf{Z}\boldsymbol{x}\|_2^2 - \frac{1}{k}\|\mathbf{Z}\boldsymbol{x}\|_2^2$$

$$\geq \frac{1}{2k}\|\mathbf{A}\boldsymbol{x}\|_2^2 - \frac{1}{k}\|\underset{\sim}{\mathbf{Z}}\boldsymbol{x}\|_2^2$$

$$\geq 1/2(1-\varepsilon) - \sigma_{\min}^2(1+\varepsilon_0)$$

$$\geq 1/2(1-\varepsilon) - \sigma_{\min}^2$$

$$\geq 1/2(1-\varepsilon) - \sigma_{\max}^2.$$

Finally, combining these results, we observe that replacing $\mathbf{A}$ with $\boldsymbol{M}$ maintains the JL guarantee with an increased distortion threshold, bounded by at most $\mathrm{poly}(\sigma_{\max}, 1/k)$, with high probability. $\qquad\square$

### H.3 Smoothing of the Johnson-Lindenstrauss indicator function

For the JL objective function we have the indicator function:

$$I(x_i; \mathbf{A}) = \begin{cases} 1 & \text{if } \left| \|\mathbf{A}x_i\|^2 - 1 \right| \geq \varepsilon \\ 0 & \text{if } \left| \|\mathbf{A}x_i\|^2 - 1 \right| < \varepsilon \end{cases}$$

and we define a smoothed version of it:

$$\tilde{I}(x_i; \mathbf{A}) = \begin{cases} 0, & \text{if } \left| \|\mathbf{A}x_i\|^2 - 1 \right| \leq \varepsilon \\ \frac{2}{\varepsilon_1^3}(\left| \|\mathbf{A}x_i\|^2 - 1 \right| - \varepsilon)^3, & \text{if } \varepsilon < \left| \|\mathbf{A}x_i\|^2 - 1 \right| \leq \varepsilon + \frac{\varepsilon_1}{2} \\ 1 - \frac{2}{\varepsilon_1^3}(\varepsilon + \varepsilon_1 - \left| \|\mathbf{A}x_i\|^2 - 1 \right|)^3, & \text{if } \varepsilon + \frac{\varepsilon_1}{2} < \left| \|\mathbf{A}x_i\|^2 - 1 \right| < \varepsilon + \varepsilon_1 \\ 1, & \text{if } \left| \|\mathbf{A}x_i\|^2 - 1 \right| \geq \varepsilon + \varepsilon_1 \end{cases}$$

for a small value $\varepsilon_1$. This smoothed indicator has gradient Lipschitz constant $L = \mathcal{O}(1/\varepsilon_1^2)$, and Hessian Lipschitz constant $K = \mathcal{O}(1/\varepsilon_1^3)$ We define the $\varepsilon_1$-smoothed version of the objective function in Equation 26, that is:

$$\tilde{f}\left(\boldsymbol{\Sigma}^{1/2}, \boldsymbol{\mu}\right) \equiv \tilde{f}(\mathbf{A}) = \mathbb{E}[\tilde{I}(x_i; \mathbf{A})].$$

and the regularized version of it, that is:

$$\hat{f}\left(\boldsymbol{\Sigma}^{1/2}, \boldsymbol{\mu}\right) \equiv \hat{f}(\mathbf{A}) = \mathbb{E}[\tilde{I}(x_i; \mathbf{A})] + \frac{\|\boldsymbol{\Sigma}^{1/2}\|}{2kd}. \tag{27}$$

By assumption we have that $1/(3n) > \mathbb{E}[I(x_i; \mathbf{A})] \geq \mathbb{E}[\tilde{I}(x_i; \mathbf{A})]$.

We also have that

$$\mathbb{E}[I(x_i; \mathbf{A})] \leq \mathbb{E}[\tilde{I}(x_i; \mathbf{A})] + \Pr(\varepsilon < \left| \|\mathbf{A}x_i\|^2 - 1 \right| < \varepsilon + \varepsilon_1).$$

Thus,

$$\sum_{i=1}^{n} \mathbb{E}[I(x_i; \mathbf{A})] \leq \sum_{i=1}^{n} \mathbb{E}[\tilde{I}(x_i; \mathbf{A})] + \sum_{i=1}^{n} \Pr(\varepsilon < \left| \|\mathbf{A}x_i\|^2 - 1 \right| < \varepsilon + \varepsilon_1).$$

We will show that when $\mathbf{A}$ has small variance (for appropriately chosen small $\rho$), the $\mathbb{E}[\tilde{I}(x_i; \mathbf{A})]$ becomes smaller than $\delta_1/n$.

We have that $\mathbb{E}[\tilde{I}(x_i; \mathbf{A})] \leq \Pr(\left| \|\mathbf{A}x_i\|^2 - 1 \right| > \varepsilon + \varepsilon_1) + \Pr(\varepsilon < \left| \|\mathbf{A}x_i\|^2 - 1 \right| < \varepsilon + \varepsilon_1)$. We assume that we have reached a point for which we have $\mathbf{A} \sim \mathcal{N}(\boldsymbol{M}, \boldsymbol{\Sigma})$. This means that $\left| \|\mathbf{A}x_i\|^2 - 1 \right|$ follows a non-central chi-squared distribution. From subgaussian properties we have:

For $t > 0$:
$$\Pr(X - \mu \geq t) \leq \exp\left(-\frac{t^2}{2\sigma^2}\right).$$

For $t < 0$:
$$\Pr(X - \mu \leq t) \leq \exp\left(-\frac{t^2}{2\sigma^2}\right).$$

We have $\mathrm{Var}\left(\left| \|\mathbf{A}x_i\|^2 - 1 \right|\right) \leq 2k\sigma_{\max}^4 + 4\sigma_{\max}^2 \sum_{i=1}^{k} \mu_i^2 := V_i$, where $\mu_i = \sum_{j=1}^{d} \mu_{ij} x_j$. Choose $t = \varepsilon + \varepsilon_1$ and if $t > \mathbb{E}[\left| \|\mathbf{A}x_i\|^2 - 1 \right|]$ we have that:

$$\Pr(\left|\|\mathbf{A}x_i\|^2 - 1\right| - \mathbb{E}[\left|\|\mathbf{A}x_i\|^2 - 1\right|] \geq t - \left|\|\mathbf{A}x_i\|^2 - 1\right|) \leq \exp\left(-\frac{(t - \mathbb{E}[\left|\|\mathbf{A}x_i\|^2 - 1\right|])^2}{2\mathrm{Var}\left(\left|\|\mathbf{A}x\|^2 - 1\right|\right)}\right)$$
$$\leq \exp\left(-\frac{(t - \mathbb{E}[\left|\|\mathbf{A}x_i\|^2 - 1\right|])^2}{2V_i}\right).$$

Otherwise if $t < \mathbb{E}[\left|\|\mathbf{A}x_i\|^2 - 1\right|]$ we have that:

$$\Pr(\left|\|\mathbf{A}x_i\|^2 - 1\right| - \mathbb{E}\left[\left|\|\mathbf{A}x_i\|^2 - 1\right|\right] \leq t - \left|\|\mathbf{A}x_i\|^2 - 1\right|)$$
$$\leq \exp\left(-\frac{\left(t - \mathbb{E}\left[\left|\|\mathbf{A}x_i\|^2 - 1\right|\right]\right)^2}{2V_i}\right). \qquad (28)$$

This implies that

$$\Pr(\left|\|\mathbf{A}x_i\|^2 - 1\right| - \mathbb{E}[\left|\|\mathbf{A}x_i\|^2 - 1\right|] \geq t - \left|\|\mathbf{A}x_i\|^2 - 1\right|) \geq 1 - \exp\left(-\frac{(t - \mathbb{E}[\left|\|\mathbf{A}x_i\|^2 - 1\right|])^2}{2V_i}\right)$$
$$\geq 1 - \frac{2V_i}{(t - \mathbb{E}[\left|\|\mathbf{A}x_i\|^2 - 1\right|])^2}.$$

Since $\mathbb{E}[\tilde{I}(x_i; \mathbf{A})] < 1/(3n)$ we have that the 2nd case where $t < \mathbb{E}[\left|\|\mathbf{A}x_i\|^2 - 1\right|]$ is rejected. We also have that for $V_i \leq -\frac{(\varepsilon + \varepsilon_1 - \mathbb{E}[\left|\|\mathbf{A}x_i\|^2 - 1\right|])^2}{\log(\delta_1/(2n))}$ the probability in Equation 28 is bounded by $\delta_1/(2n)$.

Similarly we have,

$$\Pr(\varepsilon < \left|\|\mathbf{A}x_i\|^2 - 1\right| < \varepsilon + \varepsilon_1) = \Pr(\left|\|\mathbf{A}x_i\|^2 - 1\right| < \varepsilon + \varepsilon_1) - \Pr(\left|\|\mathbf{A}x_i\|^2 - 1\right| < \varepsilon)$$
$$= \Pr(\left|\|\mathbf{A}x_i\|^2 - 1\right| > \varepsilon) - \Pr(\left|\|\mathbf{A}x_i\|^2 - 1\right| > \varepsilon + \varepsilon_1).$$

This implies that

$$\Pr(\varepsilon < \left|\|\mathbf{A}x_i\|^2 - 1\right| < \varepsilon + \varepsilon_1) \leq \Pr(\left|\|\mathbf{A}x_i\|^2 - 1\right| > \varepsilon).$$

Similarly to before we can get that:

$$\Pr(\left|\|\mathbf{A}x_i\|^2 - 1\right| > \varepsilon) \leq \exp\left(-\frac{(\varepsilon - \mathbb{E}[\left|\|\mathbf{A}x_i\|^2 - 1\right|])^2}{2V_i}\right). \qquad (29)$$

Therefore we have that for $V_i \leq -\frac{(\varepsilon - \mathbb{E}[\left|\|\mathbf{A}x_i\|^2 - 1\right|])^2}{\log(\delta_1/(2n))}$ the probability in Equation 29 is bounded by $\delta_1/(2n)$.

This means that choosing, $V_i \leq \min\left\{-\frac{(\varepsilon - \mathbb{E}[\left|\|\mathbf{A}x_i\|^2 - 1\right|])^2}{\log(\delta_1/(2n))}, -\frac{(\varepsilon + \varepsilon_1 - \mathbb{E}[\left|\|\mathbf{A}x_i\|^2 - 1\right|])^2}{\log(\delta_1/(2n))}\right\} = -\frac{(\varepsilon - \mathbb{E}[\left|\|\mathbf{A}x_i\|^2 - 1\right|])^2}{\log(\delta_1/(2n))}$ and overall, if we choose $V = \max\{V_1, \dots, V_n\}$ to satisfy all inequalities we get:

$$\sum_{i=1}^n \mathbb{E}[I(x_i; \mathbf{A})] \leq \sum_{i=1}^n \mathbb{E}[\tilde{I}(x_i; \mathbf{A})] + \sum_{i=1}^n \Pr(\varepsilon < \left|\|\mathbf{A}x_i\|^2 - 1\right| < \varepsilon + \varepsilon_1) < \delta_1.$$

Denote $M_i := \sum_{l=1}^{d} \mu_{i,l}$ and $C_1 := \min_{i=1,\dots,n} -\frac{(\varepsilon - \mathbb{E}[|\|\mathbf{A}x_i\|^2 - 1|])^2}{\log(\delta_1/(2n))}$. Then to get $V \leq C_1$, we need $\sigma_{\max}^2 \leq \min_{i=1,\dots,n} \left\{ \frac{-2M_i + \sqrt{4M_i^2 + kC_1}}{k} \right\} =: C_2$.

Choose $\delta_1 < 5/6$ and $\rho < \Delta\sqrt{C_2}$. Thus reaching a $\rho$-SOSP for $\mathbb{E}[\tilde{I}(x_i; \mathbf{A})]$ has returned a random matrix $\mathbf{A}$ that satisfies the JL guarantee with probability at least $1/6$. We will use this in the proof of Theorem 5.5 in the next section.

## H.4 Proof of Theorem 5.5

**Theorem.** Let $n$ be unit vectors in $\mathbb{R}^d$, $k$ be the target dimension, $\epsilon$ be a smoothing parameter and $\Delta > 0$ be an accuracy parameter. For any $\varepsilon \geq C\sqrt{\log n/k}$, where $C$ is a sufficiently large constant, initialize $\mathbf{M} = \mathbf{0}$ and $\mathbf{\Sigma} = \mathbf{I}_{kd}$ and run Algorithm 1 to optimize the $\varepsilon_1$-smoothed version of the objective function in Equation 9 (see Equation 27) using the regularization parameter $\lambda = \frac{\sqrt{\rho/\epsilon^3} + \Delta}{2}$. After $T = \mathcal{O}\left(\text{poly}\left(n, k, d, \log(\delta), \Delta^{-1}\right)\right)$ iterations, this returns a matrix $\mathbf{M}$ that satisfies the JL guarantee with distortion at most $\mathcal{O}(\varepsilon)$, with probability $1 - \delta$.

**Proof.** To ensure good performance, we choose the dimension $k$ such that the probability of any individual distortion constraint being violated is no more than $1/(3n)$. This choice is critical, particularly at the initialization point $(\mathbf{I}_{kd \times kd}, \mathbf{0})$, where the regularization term contributes exactly $1/2$. Under this setting, the objective function in Equation 26 satisfies:

$$f(\mathbf{I}_{kd \times kd}, \mathbf{0}) < \frac{n}{3n} + \frac{1}{2} = \frac{1}{3} + \frac{1}{2} < \frac{5}{6}.$$

This observation implies that, if we follow a monotonically decreasing path of the objective and converge to a deterministic solution, specifically one where $\mathbf{\Sigma}^{1/2} = \mathbf{0}$, the only feasible outcome is that each term in the summation becomes zero. Consequently, the distortion probability in Equation 26 will converge to zero.

However, to use our key Lemma we need to use a smoothed version of the indicator function. The smoothed objective function in Equation 27 is both gradient and Hessian Lipschitz continuous. Let $L = \mathcal{O}\left(\frac{1}{\varepsilon^2}\right), K = \mathcal{O}\left(\frac{1}{\varepsilon^3}\right)$, be the gradient and Hessian Lipschitz constants, respectively.

Choose $\delta_1 < 5/6$, $\Delta > 0$, and $\lambda = \frac{\sqrt{K\rho} + \Delta}{2} = \frac{\sqrt{\rho/\varepsilon^3} + \Delta}{2}$ and using Lemma 3.1, we get that any $\rho$-SOSP gives a matrix $\mathbf{\Sigma}^{1/2}$, that satisfies $\|\mathbf{\Sigma}^{1/2}\|_F \leq \frac{\rho}{\Delta}$.

Denote $M_i := \sum_{l=1}^{d} \mu_{i,l}$ and $C_1 := \min_{i=1,\dots,n} \left\{ -\frac{(\varepsilon - \mathbb{E}[|\|\mathbf{A}x_i\|^2 - 1|])^2}{\log(\delta_1/(2n))} \right\}$. Then to get $V \leq C_1$, we need $\sigma_{\max}^2 \leq \min_{i=1,\dots,n} \left\{ \frac{-2M_i + \sqrt{4M_i^2 + kC_1}}{k} \right\} =: C_2$.

Choose $\rho < \Delta\sqrt{C_2}$, then running Algorithm 1 with step size $\mathcal{O}(\varepsilon^2)$ for:

$$T = \mathcal{O}\left( \frac{L}{\rho^2} \log^4(d) - \frac{L}{\rho^2} \log^4(\delta) \right) = \mathcal{O}\left(\text{poly}\left(n, k, d, \log(\delta), \Delta^{-1}\right)\right),$$

returns a random matrix with $\sigma_{\max}^2 \leq \frac{\rho^2}{\Delta^2}$, that satisfies the JL guarantee with distortion $\varepsilon$ with probability at least $1/6$, with high probability. Then, from Lemma H.3, we get that we can use the mean matrix $\mathbf{M}$ which will increase the distortion threshold by at most $\text{poly}\left(\sigma_{\max}, \frac{1}{k}\right) = \text{poly}\left(\frac{\rho}{\Delta}, \frac{1}{k}\right)$, meaning that it satisfies the JL guarantee with distortion at most $\mathcal{O}(\varepsilon)$. $\square$

## H.5 Experiments in Johnson-Lindenstrauss

In this experiment, we aim to minimize the distortion introduced by random linear projections in the JL framework. A batch-based variational model is trained using the gradient-based method; SGD, which is sufficient for this task, to produce random matrices $\mathbf{A} \sim \mathcal{N}(\mathbf{M}, \mathbf{\Sigma}) \in \mathbb{R}^{k \times d}$ (with

$k = 30$ and $d = 500$) that minimize the maximum distortion when applied to a normalized dataset of $n = 100$ samples, each with $d = 500$ dimensions. Unlike traditional JL embeddings that rely on random Gaussian matrices, our approach optimizes the parameters $(\boldsymbol{M}, \boldsymbol{\Sigma})$ of a distribution over projection matrices using the Adam optimizer (Kingma & Ba, 2015) in order to minimize the worst-case distortion. We used a batch size of 20, a learning rate of 0.01, over a maximum of 5000 iterations, and early stopping is triggered if the distortion falls below 0.01. To track how the distortions evolve with our method, we sample from the current mean matrix and variance at each iteration and then calculate the resulting distortion.

Figure 6 shows the evolution of the maximum distortion throughout training, demonstrating a steady decrease. Over time, our method significantly outperforms both the average and minimum distortions obtained from standard Gaussian matrices over 1000 trials. Specifically, our learned projection achieves near-zero distortion, compared to typical random projections that yield average and minimum distortions around 1 and 0.6, respectively. Figure 7 illustrates the evolution of the maximum variance $\sigma^2$, which converges toward zero during training. This indicates that the model is refining its uncertainty and collapsing toward a deterministic, low-distortion projection. These findings suggest that structured embeddings with far lower distortion than those from conventional random constructions do exist, and that such embeddings can be effectively discovered via gradient-based optimization.

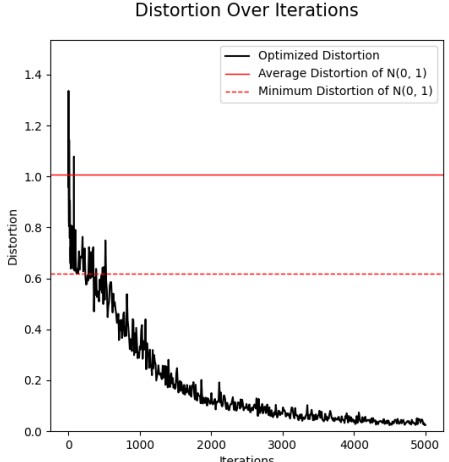

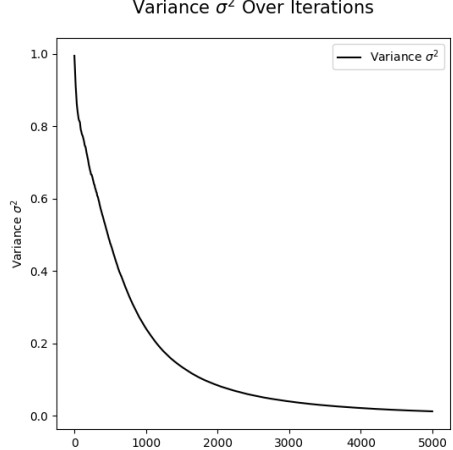

Figure 6: Evolution of the optimized distortion over iterations.

Figure 7: Evolution of maximum $\sigma^2$ over iterations.

