# OpenReview forum: "A Derandomization Framework for Structure Discovery: Applications in Neural Networks and Beyond"
_ICLR.cc/2026/Conference — ICLR 2026 Poster_

### Official Review · Reviewer_F4fe · 2025-10-17

**Soundness:** 3
**Presentation:** 2
**Contribution:** 3
**Rating:** 6
**Confidence:** 3

**Summary:**

In this paper, the authors prove a de-randomization lemma which reveals information about the loss landscape of problems that fit into a certain structural framework. The lemma essentially says that when trying to optimize a smooth function over a matrix W, such that W is multiplied by a standard normal vector and passed as an argument to the smooth function, than any candidate solution W* which is approximately second order stationary (which includes all local minima and potentially some saddle points) is close to 0, with the proximity depending on how close the point is to being second order stationary.

This lemma is then demonstrated on 3 diverse problems of interest including:
1. proving a statement about training dynamics for neural networks discovering low dimensional structure.
2. a method for derandomizing the rounding procedure which is applied to the Semi definite program solution of a continuous version of the max cut problem
3. a method of finding deterministic JL embedding matrices.

The lemma provides an interesting insight into the loss landscape for a certain class of functions. The examples demonstrate an interesting and diverse range of applications which have a common structure.

While interesting, I think demonstrating the importance of the main lemma hinges on the examples. The first example which received the majority of the attention in the paper is solid and interesting but feels somewhat marginal in terms of extending  the result of Mousavi-Hosseini et al., 2023. I do think the other two examples are very interesting but I do not totally understand the impact of the lemma in regards to the problems it is applied to. If this was highlighted more clearly and I can be convinced of the significance this lemma brings to the MaxCut and JL applications, I would think the paper is much stronger.

**Strengths:**

The various applications are interesting as they consider very different but high profile problems which are of interest to many different communities.

The general insight into the loss landsape is also interesting and I agree with the authors that others may find new applications of this lemma to different problems.

**Weaknesses:**

It seems as though for the L-smooth function who's expectation we want to optimize, the sample complexity for finding a solution with perturbed gradient descent is growing in L. When applying the lemma to optimize probabilities (written as expected indicators) the smooth approximation to the indicators requires larger L for better approximation. In these cases there is an obvious tradeoff between sample complexity and accuracy. I think this is clear for the max-cut theorem but is sort of hidden in the JL example and perhaps this should be made more clear.

To follow up on the point above, given the application of the lemma to these two problems, it would be interesting to have some insight into how quality of approximation of the indicator affects quality of the final solution.

It would be nice to have a bit more context into the final two applications in terms how effective the proposed de-randomization problems are. You suspect that you have shared the first optimization-based approach for
derandomizing the Goemans-Williamson algorithm, but have not commented on how this approach compares to the other mentioned methods of  of conditional expectations, small-bias spaces, or explicit pseudorandom constructions. For JL you mentioned that you match the SOTA results but also later that actually recover another method and there is no real comparison between your or other methods (See the first question in the 'Questions' section).

I think the connections of the structure used in your lemma and these different applications are interesting. It is not totally clear to me however what exactly applying your lemma to these problems is contributing to the study of these problems. Is it new methods for solving these problems? Or contributing to the theoretical understandings of these problems or something else? If it is new methods I think a better comparison is warranted.

**Questions:**

For JL how exactly does your method differ from what others have already done? Or is the contribution of this example to demonstrate that your lemma can be applied and hence provide insight into the loss landscape for this problem?

A small note: I believe in appendix C (line 903) the definition of $\ell'$ should include the $\| W_\|\|^2_F$ not $\| W_\perp\|^2_F$ right?

---

> ### Author Response · Authors · 2025-11-20
>
> We thank the Reviewer for their time and thoughtful feedback, which has provided valuable guidance in clarifying our contributions. We have carefully considered each comment, and our detailed responses are provided below.
>
> $\newline$
>
> $ \textbf{Q1: While interesting, I think demonatrating the importance of [...]. +  I think the connections of the structure [...].} $
>
>
> A1: Regarding the MAXCUT and JL examples, we are glad the Reviewer appreciates the value in these applications. Our goal is not to propose new algorithms, but to provide a theoretical framework clarifying when and how randomness can be removed in such problems, leading to improved solutions. Specifically, whenever a randomized algorithm can be expressed in the general form of Equation (2), as in the JL and MAXCUT settings, our lemma guarantees the existence of a deterministic equivalent with performance at least as good (up to the small error from using approximate rather than exact SOSPs). In this sense, our contribution is a general theoretical tool: once an objective fits the structure of Equation (2), our framework can systematically reduce its randomness without any complicated problem-specific analysis or ad hoc tricks that are typically required in this literature (see references of our answer A2 to Reviewer uty6).
>
>
> Regarding the neural network example, we believe our work goes beyond a marginal extension of Mousavi-Hosseini et al., 2023. In particular, we resolve an open question left in that work regarding the necessity of an unnaturally large regularization, as we show that the same guarantees hold under minimal regularization. Moreover, our framework allows trainable biases and arbitrary network depth, while Mousavi-Hosseini et al., 2023 (and most prior works) were limited to two-layer networks with fixed random biases.
>
> $\newline$
>
> $ \textbf{Q2: It seems as though for the L-smooth function [...]. + To follow up on the point above [...].} $
>
> A2: We agree that the tradeoff between sample complexity and approximation accuracy could be made clearer, and we will clarify this explicitly for the JL case as well.
> The use of smooth approximations is primarily a technical detail for the theoretical guarantees of the lemma, and is standard in the literature. In practice, we have not observed any noticeable difference in performance when replacing non-smooth functions with their smooth approximation counterparts. We will make this empirical robustness clearer in the camera-ready version.
>
> $\newline$
>
> $ \textbf{Q3: It would be nice to have a bit more context [...]. + For JL how exactly does your method differ [...].} $
>
> A3: Our approach provides a systematic way to translate the original problem into a continuous optimization problem. This translation allows us to leverage the extensive theory and guarantees of smooth nonconvex optimization, including well-studied tools such as second-order stationarity. In this sense, our method is not meant to compete directly with classical derandomization techniques such as those you mentioned.
>
> Importantly, our method matches the state-of-the-art in terms of approximation quality for the problems we considered while avoiding the need to “open the box” and examine problem-specific constructions. Its generality is highlighted by its applicability across three very different domains: structure discovery in neural networks, combinatorial optimization (MAXCUT), and dimensionality reduction (Johnson-Lindenstrauss).
>
>    In contrast, prior JL work is largely problem-specific, and its techniques cannot be easily adapted and transferred to other problems. Our framework, by contrast, provides a flexible tool that can be applied broadly to many problems, essentially any problem that can be written in the very general form of Equation (2). Beyond this, as you correctly identify, it also offers insight into the loss landscape of problems of this form.
>
> $\newline$
>
> $\textbf{Q4:  A small note: I believe in appendix C [...].}$
>
> A4: You are right regarding the typo in line 903; we corrected it.

---

> > ### Comment · Reviewer_F4fe · 2025-11-21
> >
> > Thank you for the response. This makes some of your intentions of this work more clear. I will maintain my positive score.

---

> > > ### Author Response · Authors · 2025-11-21
> > >
> > > Thank you for your answer. If you have any further comments or suggestions, we are happy to consider them.

---

### Official Review · Reviewer_uty6 · 2025-11-01

**Soundness:** 3
**Presentation:** 2
**Contribution:** 3
**Rating:** 6
**Confidence:** 2

**Summary:**

This paper develops a general derandomization framework for structure discovery in neural networks and other domains. Building on the prior work of (Mousavi-Hosseini et al., 2023), which showed that SGD can lead to low-rank structure in the first layer of a two-layer neural network under strong regularization, this paper extend the result to a broader setting with trainable bias parameters, general smooth loss functions and optimizers. The core contribution is a key derandomization lemma stating that optimizing a general regularized loss in expectation form converges to the origin point, under mild conditions. The authors also apply the same lemma to derive derandomized algorithms for the MAXCUT problem and Johnson–Lindenstrauss embeddings.

**Strengths:**

1. The authors propose a new derandomization lemma for analyzing feature learning behaviour of neural networks. The obtained results extend and generalize the analysis in (Mousavi-Hosseini et al., 2023).
2. The authors applied the result for derandomization in other domains including MAXCUT and JL embeddings.

**Weaknesses:**

1. The assumption of first and second order smoothness is restrictive and does apply to many practical scenarios, such as ReLU networks, forcing the authors to adopt the approximation in Section 4.2. Can this result be extended to non-smooth settings such as the original ReLU activation?
2. The applications to MAXCUT and JL embedding are actually not new since, as acknowledged by the authors, there are already known derandomized algorithms for both problems. Can this result be applied to new problems where derandomized algorithms are unknown?

Typos and other comments:

1. in line 903, the definition of $l’_{\\theta’}$ below equation (18), $\\lambda\\|W{\\bot}\\|\_{F}^2$ should be $\\lambda\\|W\_{\\|}\\|\_{F}^2$.
2. The “Summary of our contributions.” section seems to be missing.

**Questions:**

See strengths and weaknesses section.

---

> ### Author Response · Authors · 2025-11-20
>
> We sincerely thank the Reviewer for their constructive feedback and valuable suggestions. Their comments have helped us identify areas where additional clarification was needed. Below, we address each of their points in detail:
>
> $\newline$
>
>
> $\textbf{Q1: The assumption of first and second order smoothness [...].}$
>
>
> A1: We thank the Reviewer for this comment. The concept of SOSPs cannot be directly defined for non-smooth functions, so we adopt smooth approximations which is a standard approach in the literature, primarily as a technical detail to demonstrate that our framework can handle such cases as well. This allows our framework to handle non-smooth activations in a principled way, since we can approximate them arbitrarily well by smooth functions. Empirically, we observed identical results when using ReLU and its smooth variant, suggesting that the smoothing does not significantly affect outcomes. Extending the theoretical guarantees to fully non-smooth activations is an interesting open direction for future work.
>
> $\newline$
>
>
> $\textbf{Q2: The applications to MAXCUT and JL embedding [...].}$
>
> A2: We agree that the derandomizations of MAXCUT and the JL lemma are well studied, with many existing constructions. Our contribution is not the fact of derandomization itself, but that such results can be achieved via simple gradient-based optimization, without relying on explicit combinatorial constructions or pseudorandom generators. This provides a different perspective to the extensive literature on derandomization, which has traditionally focused on increasingly sophisticated constructions, see, for example, [1,2,3] and references therein, which are typically considerably more complex than our optimization-based approach.
>
> More generally, for any objective that can be written in the form of Equation (2), our lemma could be applied.
>
> $\newline$
>
>
>
> $\textbf{Q3: In line 903, the definition of [...].}$
>
> A3: Thanks for the observation, you are right, this was a typo. We corrected it in the updated version.
>
> $\newline$
>
>
> $\textbf{Q4: The “Summary of our contributions.” [...].}$
>
> A4: In lines 133-134 we have written a paragraph "Summary of our contributions.” We can change this into a section in the camera-ready version.
>
> $\newline$
>
>
> [1]: Kane, D.M. and Nelson, J., 2010. A derandomized sparse Johnson-Lindenstrauss transform. arXiv preprint arXiv:1006.3585.
>
>
> [2]: Kaplan, E., Naor, M. and Reingold, O., 2009. Derandomized constructions of k-wise (almost) independent permutations. Algorithmica, 55(1), pp.113-133.
>
> [3]: Meka, R. and Zuckerman, D., 2010, June. Pseudorandom generators for polynomial threshold functions. In Proceedings of the Forty-second ACM Symposium on Theory of Computing (pp. 427-436).

---

> ### Comment · Reviewer_uty6 · 2025-11-28
>
> Thank the authors for their detailed response. I maintain my score.

---

### Official Review · Reviewer_n2cY · 2025-11-04

**Soundness:** 3
**Presentation:** 3
**Contribution:** 2
**Rating:** 4
**Confidence:** 4

**Summary:**

This manuscript studies representation learning in neural networks in a multi-index setting. In particular, the authors show that, using the definition of approximate second-order stationarity, the result in (Mousavi-Hosseini et al., 2023) can be established under more general conditions. They provide algorithms that find approximately second-order stationary points under smoothness assumptions and also suggest alternative uses of these algorithms for MAX-cut and embedding finding.

**Strengths:**

- The paper extends earlier result (Mousavi-Hosseini et al., 2023) to a broader setting.

**Weaknesses:**

- The paper does not guarantee learning the teacher directions; rather, it shows that the component of the student weights in the subspace orthogonal to the teacher directions vanishes. However, this does not guarantee recovery of the teacher directions. For example, consider the setting where the teacher is $y = \mathrm{He}_4(\langle \theta, x\rangle) + \epsilon$ for some unit vector $\theta$, and the student is $\hat y = \mathrm{He}_2(\langle w, x\rangle)$, $w \in \mathbb{R}^d$, with $x \sim N(0, I_d)$ and square loss. Then $0$ will be an approximate stationary point for any $\rho > 0$, even though it is a saddle point. In this sense, the result does not guarantee learning.

- The example above relies on the orthogonality of the Hermite polynomials $\mathrm{He}_4$ and $\mathrm{He}_2$. Hermite decompositions in multi-index settings have been studied extensively since (Mousavi-Hosseini et al., 2023), and there is now a rich literature that discusses the sample complexity of recovering teacher model with first-order algorithms based on its Hermite decomposition in Gaussian space [1,2,3]. The manuscript does not compare its results to this literature. What improvement does this paper provide? The manuscript would benefit from this discussion.


[1] Gérard Ben Arous, Reza Gheissari and Aukosh Jagannath. “Online stochastic gradient descent on non-convex losses from high-dimensional inference.” J. Mach. Learn. Res. 22 (2020): 106:1-106:51.

[2] Alex Damian, Loucas Pillaud-Vivien, Jason D. Lee and Joan Bruna. “Computational-Statistical Gaps in Gaussian Single-Index Models.” ArXiv abs/2403.05529 (2024): n. pag.

[3] Jason D. Lee, Kazusato Oko, Taiji Suzuki and Denny Wu. “Neural network learns low-dimensional polynomials with SGD near the information-theoretic limit.” ArXiv abs/2406.01581 (2024): n. pag.

**Questions:**

- In Eq. (3), $U$ is not bold; I believe this is a typo.

- Following my points above, in line 260 the authors claim:
  “Our goal is to show that the perpendicular component $W_{\perp}$ converges to zero, implying that the first-layer weight matrix $W$ converges to a rank-$k$ matrix,”
  which is incorrect. This statement should be revised.

---

> ### Author Response · Authors · 2025-11-20
>
> We appreciate the Reviewer's time and effort in reviewing our work. The feedback has highlighted important aspects for clarification, and we hope that further discussion will help address any remaining concerns. Below, we provide their responses to each point:
>
> $\newline$
>
> $\textbf{Q1: The paper does not guarantee learning the teacher directions; [...].}$
>
> A1: The established connection between structure discovery and generalization (Mousavi-Hosseini et al., 2023) does not always imply learning as explained already in that manuscript. Typically you need much stronger assumptions, see Theorem 4 in that paper (e.g. monotone link function, random frozen biases, single-index model etc.).
>
> Because the express focus of our work is on improving and better understanding the principles underlying structure discovery, we do not explicitly study the learning implications. Instead, our theorem isolates the mechanism by which optimization methods keep the iterates within the teacher subspace, and it strengthens these structure discovery results by loosening the assumptions, e.g. regarding regularization.
>
> $\newline$
>
> $\textbf{Q2: The example above relies on the orthogonality of the Hermite polynomials [...].}$
>
>
> A2: Thank you for giving us the opportunity to clarify this important topic and to improve the discussion in our manuscript (see Appendix A). The cited works [1,2,3] analyze single-index models, proving guarantees for when SGD or SQ-type algorithms can succeed under that specific structure. To the best of our knowledge, most existing learning results in this literature also focus exclusively on the single-index setting. In contrast, as we mention in the abstract and the introduction, our work takes a distinct perspective: we study the optimization landscape of general Gaussian-expectation objectives and establish a deterministic derandomization lemma that guarantees strong structural properties under minimal regularization without assuming polynomial teachers or single-index structure. Our contribution, therefore, concerns the geometry of the objective, rather than sample complexity within the simpler single-index framework, and is orthogonal and complementary to the algorithmic analyses of [1,2,3]. In summary, our goal is not to tighten sample complexity bounds, but to offer a new geometric characterization of how optimization methods confine learning to the teacher subspace under very minimal assumptions, an aspect not captured by prior works.
>
> $\newline$
>
> $\textbf{Q3: In Eq. (3) [...].}$
>
> A3: You are right about the typo, we changed it.
>
> $\newline$
>
> $\textbf{Q4: Following my points above, in line 260 the authors claim [...].}$
>
> A4: Thanks for pointing that out. We agree that this statement can be misleading. What we aim to show is that the perpendicular component $\textbf{W}_{\perp}$ vanishes, which implies that the first-layer weight matrix $\textbf{W}$ lies entirely in the teacher subspace. We have updated the manuscript (lines 260-262) to reflect this more precise statement.

---

> > ### Comment · Reviewer_n2cY · 2025-11-26
> >
> > I thank the authors for their response. I would like to respond to their reply to Q2 and reiterate some of my concerns about the comparison with prior work.
> >
> > * **Multi-index setting.** I respectfully disagree with the authors’ suggestion that the prior work focuses *exclusively* on the single-index setting. There are works [1–5] that analyze the multi-index setting and guarantee sample complexity for recovering the teacher direction/subspace. The current result in the manuscript, on the other hand, is a population-level result (so it does not provide sample complexity), and even in the population case it is not clear whether it guarantees full recovery of the teacher.
> >
> > * **Minimal assumptions and geometric characterization.** One aspect in which this manuscript improves upon the line of work I mentioned above is allowing a mismatch in the link functions of the teacher and student networks, which is admittedly an important problem. There is some existing work that studies this mismatched setting based on Hermite expansions of the teacher and student networks, in the single-index setting [6, 7] and in the multi-index setting [8]. One arguably weak part of the present work is that it is not obvious from the statement of the main result when it becomes vacuous and when it actually guarantees learning. In that regard, a discussion of when the manuscript goes beyond existing results is missing. I would appreciate further clarification on when the guarantees are non-vacuous and how they compare to these prior results before considering updating my score.
> >
> > [1] Damian, A., Lee, J.D., & Soltanolkotabi, M. (2022). Neural Networks can Learn Representations with Gradient Descent. ArXiv, abs/2206.15144.\
> > [2] Abbe, E., Boix-Adserà, E., & Misiakiewicz, T. (2023). SGD learning on neural networks: leap complexity and saddle-to-saddle dynamics. ArXiv, abs/2302.11055.\
> > [3] Ben Arous, G., Gerbelot, C., & Piccolo, V. (2024). Stochastic gradient descent in high dimensions for multi-spiked tensor PCA. ArXiv, abs/2410.18162.\
> > [4] Ben Arous, G., Erdogdu, M.A., Vural, N., & Wu, D. (2025). Learning quadratic neural networks in high dimensions: SGD dynamics and scaling laws.\
> > [5] Ren, Y., Nichani, E., Wu, D., & Lee, J.D. (2025). Emergence and scaling laws in SGD learning of shallow neural networks. ArXiv, abs/2504.19983.\
> > [6] Bietti, A., Bruna, J., Sanford, C., & Song, M.J. (2022). Learning Single-Index Models with Shallow Neural Networks. ArXiv, abs/2210.15651.\
> > [7] Pillaud-Vivien, L., & Schertzer, A. (2025). Joint Learning in the Gaussian Single Index Model. ArXiv, abs/2505.21336.\
> > [8] Bietti, A., Bruna, J., & Pillaud-Vivien, L. (2023). On Learning Gaussian Multi-index Models with Gradient Flow. ArXiv, abs/2310.19793.

---

> ### Author Response · Authors · 2025-11-28
>
> We thank the Reviewer for their answer and for engaging with our response, and we hope this clarifies their questions.
>
> $\textbf{Q1: Multi-index setting. [...].}$
>
> A1: We agree with the reviewer's comment. Our intention was simply to note that, to the best of our knowledge, most existing learning results in this literature focus on the single-index setting, which is typically simpler to analyze. We did not mean to imply that multi-index analyses are nonexistent. In fact, as we mention in lines 806–807 in the Appendix, we explicitly cite prior work on multi-index models.
>
>
> Regarding the comment on the finite-sample case, we view the problem from a slightly different perspective: our main focus is on showing that a $\rho-$SOSP solution reveals structure, rather than providing sample complexity. However, we do cover the finite-sample case as there exist algorithms that yield $\rho-$SOSP solutions. This discussion is already mentioned in the main body (lines 240–243).
>
>
> $\textbf{Q2: Minimal assumptions and geometric characterization. [...].}$
>
> A2: We would like to clarify that our manuscript does not make any claims about learning, as these are outside the scope of our work. Our contributions are strictly focused on the structural properties of the optimization landscape and on how these properties confine first-layer weights to the teacher subspace under very minimal assumptions.
>
> While our results have an intuitive connection to learning, formally establishing such a connection would require analysis under stronger assumptions.

---

### Official Review · Reviewer_cEJx · 2025-11-07

**Soundness:** 4
**Presentation:** 3
**Contribution:** 3
**Rating:** 8
**Confidence:** 3

**Summary:**

Consider the problem of optimizing any function of the form $\mathbb{E}[g(Wx + b)] + \lambda \\|W\\|^2_F$, where $x$ is drawn from a standard Gaussian, with respect to parameters $W, b$. The main result of this paper states that any second-order stationary point (SOSP) of this objective must have $W = 0$ (and an approximate version holds for approximate SOSPs). This has several applications:
1. Neural network optimization: when we try to optimize the squared loss between a "teacher" network of the form $h(Ux)$ and a "student" neural network with initial layer weights $W$, then at any SOSP $W$ will have its component orthogonal to $U$ approximately vanish. This could be seen as optimization "discovering" the (typically low-rank) subspace of $U$.
2. Derandomization: this lemma can be cleverly applied to certain randomized algorithms (such as MAX-CUT or Johnson-Lindenstrauss) in obtain a deterministic solution that (nearly) matches the performance of the randomized one.
The proof of the main lemma is simple and relies crucially on Stein's lemma for Gaussian random variables.

**Strengths:**

The main result of the paper is appealingly clean, general, and powerful. The applications are creative and interesting. In the case of neural network optimization, the paper recovers a "structure discovery" result which had been proved much more painstakingly (and under more restrictive assumptions) in prior work. The derandomization applications are conceptually thought-provoking and potentially of independent interest in complexity theory and randomized algorithms. The paper is mostly written quite clearly. I did not verify all the proofs in detail but the main ideas seem technically sound (and in particular the main lemma proof looks correct). Overall, this looks like a good paper.

**Weaknesses:**

A weakness of the paper is that it is stylized in certain important ways. Probably the biggest concern is that all the results rely on perfect Gaussianity of the random variables, because of the crucial use of Stein's lemma in the main result. Another concern is that the functions involved be smooth, which necessitates the use of smooth activation functions etc. Finally, all the claims require regularization and only hold for $\rho$-SOSPs with $\rho$ very small compared to the regularization strength $\lambda$. This limits the interpretation of some of the more practical applications, esp concerning neural network optimization. Indeed, in that application it is not clear that this meaningfully explains "structure discovery" in any way that illuminates what might be happening in a real world setting; rather this seems like more of a curiosity about the Gaussian. The algorithmic applications only really seem suitable for cases where the algorithm designer synthetically injects Gaussian randomness into an algorithm (as with randomized rounding and random projections), not cases with "real world" randomness. This is probably fine, but it bears noting.

**Questions:**

1. I am curious if the results in this paper extend even in an approximate way to distributions that are close to but not perfectly Gaussian (e.g. sub-Gaussian, log-concave, etc). Would be curious to hear the authors' thoughts on this and more generally the restrictiveness of the Gaussianity assumption.
2. In general throughout the paper, the authors need to be clearer about what the variables being optimized are in any given objective function. Even the term SOSP is always really "SOSP of certain optimization variables". This is sometimes but not always clear from context. In particular, in section 4, I was quite confused by the step in going from Eq 5 to Eq 6, because $W_{\parallel}$ suddenly disappears from the objective altogether. When we actually optimize the risk, we optimize all of $W$ --- i.e., really we are considering an SOSP of both $W_{\bot}$ and $W_{\parallel}$. It is also extra confusing because $\ell'$ implicitly depends on $W_{\parallel}$. I had to spend some time convincing myself that we can effectively work as if $W_{\parallel}$ were a constant equal to its SOSP value. Perhaps such manipulations are obvious to an optimization audience but I think they are somewhat subtle and deserve very clear, explicit exposition (perhaps in the appendix, if space does not permit).
3. The interpretation of the main lemma as a "derandomization" lemma deserves clearer explanation. It only really comes up in Section 5 and even there it is a bit conceptually subtle. In fact this is a good opportunity in the paper to state purely in words what the main lemma is really saying.

---

> ### Author Response · Authors · 2025-11-20
>
> We sincerely appreciate the Reviewer's insightful review of our work. Their comments have helped us address some important points, and we hope that through further discussion, we can clarify any remaining concerns. Below, we provide our responses to each point:
>
> $\newline$
>
> $ \textbf{Q1:  I am curious if the results [...].} $
>
> A1: The Gaussian data is the most common assumption in state-of-the-art theoretical studies of neural networks, particularly in teacher–student analyses [1,2,3,4]. Our work adopts this standard setting. Although we believe our results may extend to more general distributions such as sub-Gaussian ones, the corresponding analysis is considerably more delicate and is beyond the scope of this paper, indeed, this analytical difficulty is the main reason why the existing literature concentrates on the Gaussian case.
>
> $\newline$
>
>
> $ \textbf{Q2: In general throughout the paper [...].} $
>
> A2: We thank the Reviewer for the suggestion to include a detailed explanation in the appendix. We have added this clarification in Appendix "Discussion on $\rho$-SOSPs.
> The subtlety arises from enlarging the parameter set from $\theta$ to $\theta^{\prime}$ where $\theta^{\prime}$ includes the parameters coming from $\mathbf{W_{\parallel}}$. Thus, $\mathbf{W_{\parallel}}$ does not disappear, its contribution is absorbed into $\theta^{\prime}$ .
> We have also corrected a minor notational issue in Appendix C. We would be happy to expand this further if the Reviewer finds it helpful.
>
> Finally, we clarify that our analysis is based on convergence to $\rho$-SOSP with respect to all parameters, including those collected in $\theta$. That is, the approximate SOSP is taken jointly over $(\mathbf{W_{\perp}}$, $\textbf{b}$, $\theta^{\prime})$. Once such a point is reached, fixing $\theta^{\prime}$ at its SOSP value preserves $\rho$-second-order stationarity with respect to $\mathbf{W_{\perp}}$, which justifies focusing on $\mathbf{W_{\perp}}$ alone in the subsequent analysis. We have added a discussion in lines 221-223 and lines 935-941.
>
> $\newline$
>
>
> $\textbf{Q3: The interpretation of the main lemma [...].}$
>
> A3: To provide intuition on the interpretation of "derandomization", consider the ideal case where $\rho = 0$. Then the lemma implies $\mathbf{W} = 0$, which in turn means that $\mathbf{W}\mathbf{x} = 0$ and all randomness from $\mathbf{x}$ is effectively removed from the objective, i.e., the objective is fully derandomized. More generally, the lemma shows that at approximate $\rho-$SOSPs, the contribution of the random input is controlled via $\|\|\mathbf{W}\|\|_F$, providing a principled way to "derandomize" the objective while maintaining or improving its value. In the context of neural network structure discovery, this means that derandomization implies that the prediction is not affected by irrelevant directions in $\mathbf{x}$. We will clarify this interpretation in the introduction.
>
> $\newline$
>
>
> [1] Emmanuel Abbe, Enric Boix Adsera, and Theodor Misiakiewicz. “SGD learning on neural networks: leap complexity and saddle-to-saddle dynamics.” Proceedings of the Thirty-Sixth Annual Conference on Learning Theory, pages 2552–2623. PMLR, 2023.
>
> [2] Alberto Bietti, Joan Bruna, and Loucas Pillaud-Vivien. “On learning Gaussian multi-index models with gradient flow.” arXiv preprint arXiv:2310.19793, 2023.
>
> [3] Jimmy Ba, Murat Erdogdu, Taiji Suzuki, Denny Wu, and Tianzong Zhang. “Generalization of two-layer neural networks: An asymptotic viewpoint.” In International Conference on Learning Representations, 2020.
>
> [4] Alberto Bietti, Joan Bruna, Clayton Sanford, and Min Jae Song. “Learning single-index models with shallow neural networks.” Advances in Neural Information Processing Systems, vol. 35, pages 9768–9783, 2022.

---

### Author Response · Authors · 2025-11-20

We sincerely thank all the Reviewers for their time and effort in evaluating our work. Their feedback has been invaluable, and we deeply appreciate the constructive comments aimed at improving our paper. We have carefully considered and incorporated these suggestions to enhance the overall quality of our work. Please see our updated manuscript, where we have highlighted in red the changes addressing the $\rho$-SOSP discussion, Hermite polynomial literature, and various typographical corrections.

---

### Meta-Review · Area_Chair_Aiqx · 2026-01-03

**Summary:**

This paper proposes a general derandomization framework for structure discovery in neural networks, with applications to MAXCUT approximation and JL embeddings. Most reviewers recommend acceptance, with one reviewer leaning toward weak rejection. The main initial concerns focus on (i) the extensibility of the results (e.g., beyond the Gaussian setting or beyond smooth activation functions) and (ii) the significance of the paper's derandomization lemma.

Reviewer n2cY, who expresses the strongest reservations, mainly argues that it is unclear when the main result becomes vacuous versus when it obtains meaningful guarantees, and that the paper does not sufficiently clarify how its contributions go beyond existing results in terms of learning implications. In response, the authors emphasize that learning guarantees are intentionally out of scope, and that the paper focuses strictly on structural properties of the optimization landscape. While I also agree that learning-related guarantees would indeed strengthen the overall impact, this omission can reasonably be deferred to future work. Given the paper's current theoretical contributions, I believe it still warrants acceptance.

**Reviewer Concerns:**

Reviewer cEJx asks whether the analysis can be extended beyond the Gaussian setting (e.g., to sub-Gaussian distributions) and raises several clarification questions. The authors acknowledge that their results rely on the Gaussian assumption and that such extensions may be non-trivial, and they also clarify the remaining points. I consider these concerns addressed.

Reviewer uty6 questions whether the analysis can be extended beyond smooth activations (e.g., to ReLU) and requests a clearer comparison to previous derandomization results for MAXCUT and JL embeddings. The authors discuss the potential applicability of their approach to non-smooth activations, and emphasize that their derandomization results mainly focus on gradient-based optimization. The reviewer indicates that these concerns were addressed.

Reviewer F4fe raises similar questions about the significance of the derandomization lemma. After the rebuttal, this reviewer likewise notes that the concerns were addressed.

The remaining point of disagreement comes from Reviewer n2cY, who maintains that the paper lacks results on learning guarantees in its current form.

**Reviewer Scores:**

Reviewer uty6 and Reviewer F4fe explicitly state that they will maintain their positive evaluations of the paper.

Reviewer cEJx is also likely to keep the score, given that the initial evaluation is already relatively strong.

Although Reviewer n2cY engaged with the authors during the discussion and remains open to updating the score. However, it is possible that the authors' response that learning guarantees are out of scope will not fully address this reviewer's core concern, and thus consensus may not be reached. While I understand this perspective, I believe the paper, in its current form, still warrants acceptance.

---

### Decision · Program_Chairs · 2026-01-26

Accept (Poster)